# Projected changes in land carbon store over the 21[st] century: what contributions from land-use change and atmospheric nitrogen deposition?

Jaime A. Riano Sanchez[1,a], Nicolas Vuichard[1], Philippe Peylin[1]

[1] Laboratoire des Sciences du Climat et de l'Environnement, LSCE-IPSL (CEA-CNRS-UVSQ), Université Paris-Saclay 91191 Gif-sur-Yvette, France

[a] now at EthiFinance, 11 Av. Delcassé, 75008 Paris

*Correspondence to*: Nicolas Vuichard (nicolas.vuichard@lsce.ipsl.fr)

**Abstract.** Earth System Models (ESM) represent the time evolution of the biophysical (energy, water cycles) and biogeochemical (carbon cycle) components of the Earth. When used for near-future projections in the context of the Coupled Model Intercomparison Project (CMIP), they use as forcings the evolution of greenhouse gas and other pollutant concentrations and land-use changes simulated by an ensemble of Integrated Assessment Models (IAMs) for a combination of socio-economic pathways and mitigation targets (SSPs). More precisely, only one IAM output is used as representative of a single SSP. This makes the comparison of key ESM diagnostics among SSPs significantly noisy, without the capacity of disentangling SSP-driven and IAM-driven factors. In this paper, we quantify the projected change in land carbon store (CLCS) for the different SSPs with an advanced version of a land surface model embedded into IPSL-CM6 ESM. Through a set of land-only factorial simulations, we specifically aim at estimating the CLCS dispersions associated with land-use change and nitrogen deposition trajectories. We showed that the spread of the simulated change in global land carbon store induced by the uncertainty on land-use changes is slightly larger than the one associated with the uncertainty on atmospheric $CO_2$. Globally, uncertainty associated with N depositions is responsible for a spread in CLCS lower by a factor three, than the one driven by atmospheric $CO_2$ or land-use changes. Our study calls for making available additional IAM scenarios for each SSP to be used in the next CMIP exercise, in order to specifically assess the IAM-related uncertainty impacts on the carbon cycle and the climate system.

## 1 Introduction

In the framework of the Phase 6 of the Coupled Model Intercomparison Project (CMIP6), the ScenarioMIP experiments (O'Neill et al., 2016) address the near-future evolution (2015-2100) of the Earth System for a combination of socio-economic and climate policy scenarios. Five shared socio-economic pathways (SSPs) are explored (Riahi et al., 2017) with contrasted assumptions regarding the future evolution of society in terms of population growth, economic development, urbanization and

other factors. Driven by these five socio-economic pathways, an ensemble of Integrated Assessment Models (IAMs) simulate
the evolution of energy and land-use systems and the associated emissions of GHG and other pollutants. In the context of
ScenarioMIP, a selection of simulations is performed for the five socio-economic pathways with or without mitigation strategy
(baseline scenario) leading to specific radiative forcings in 2100 (O'Neill et al., 2016). As defined in O'Neill et al. (2016), we
label these eight scenarios as SSPx-y with x the selected SSP and y the 2100 radiative forcing. Here forward, we refer to these
scenarios as SSPs for simplicity. In order to be used by Earth System Models (ESM), IAMs outputs are harmonized to be
consistent with the data used for the historical period and downscaled from the IAMs large-region scale to a finer gridded one.
Harmonization and downscaling are performed for land-use (Hurtt et al., 2020) and for emissions of GHG and other
atmospheric compounds impacting climate such as ammonia or nitrogen oxides (Feng et al., 2020; Gidden et al., 2019).
Most of the CMIP6 experiments designed to assess the contemporary evolution of the Earth system have been performed in a
so-called concentration-driven mode. In such configuration, atmospheric $CO_2$ concentration ([$CO_2$]) is imposed, and fossil
$CO_2$ fuel emissions are computed a-posteriori as the remaining flux compatible with the time evolution of [$CO_2$] and the net
land-atmosphere and ocean-atmosphere $CO_2$ fluxes. Liddicoat et al. (2021) computed the compatible fossil fuel $CO_2$ emissions
deduced from the historical and ScenarioMIP experiments of nine ESMs. They showed that the multimodel mean cumulative
compatible fossil fuel $CO_2$ emissions over 1850-2100 were in closed agreement with the estimate based on observation (for
the historical period) and the IAMs (for the period 2015-2100) for the different SSPs. The absolute relative difference between
the multimodel mean and the observation/IAM-based estimate ranges from 1% (for SSP3-7.0) to 13% (for SSP1-1.9), proving
the overall good consistency between ESM and IAM carbon (C) cycle modelling. However, the model spread is large, with an
intermodel standard deviation ranging from 5% (for SSP5-8.5) to 15% (for SSP4-3.4) of the multimodel mean compatible
fossil fuel $CO_2$ emissions. This large disagreement between ESMs is primarily attributable to the land carbon response, with
an intermodel standard deviation for the land carbon store between 1850 and 2100 of the order of 67% of the multimodel
mean, while the one for the ocean carbon store does not exceed 6%.
In this context, our paper focus on the projected ESM land carbon store for the different SSPs and in particular on an additional
source of uncertainty related to the IAM forcings. Indeed, five IAMs simulated the evolution of the energy and land-use
systems and associated gas emissions for each SSP but only outputs of a single IAM per SSP have been harmonized and
downscaled to be further used as ESM inputs. These selected interpretations of SSPs are called "markers" and the other IAM
scenarios for each SSP "non-markers" (Riahi et al., 2017). While the anthropogenic $CO_2$ emission trajectories simulated by
the different IAMs for a given SSP are relatively similar (https://tntcat.iiasa.ac.at/SspDb, see also (Bauer et al., 2017) for a
specific analysis for fossil fuel emissions only), there are large inter-IAM spreads for land-use trajectories (Riahi et al., 2017;
Popp et al., 2017) but also for nitrogen (N) fertilizer usage (Sinha et al., 2019) and pollutant emissions (in particular ammonia,
https://tntcat.iiasa.ac.at/SspDb).
This selection of marker IAMs as representatives of a single SSP while the inter-IAM spread is large, makes difficult the
uncertainty analysis of key ESM diagnostics as a function of SSPs, without the capacity of disentangling SSP-driven and IAM-
driven factors (Sinha et al., 2019; Monier et al., 2018). While this difficulty gets support to the development of coupled human-

Earth system (CHES) models (Monier et al., 2018; Golaz et al., 2022) to gain in modelling consistency, this option does not facilitate the assessment of an IAM-specific uncertainty and of its impact on the ESM diagnostics.

In this paper, we quantify the projected change in land carbon store (CLCS) for the different SSPs from land-only simulations of the ORCHIDEE-v3 land surface model (LSM) (Vuichard et al., 2019) driven by climate data from the IPSL-CM6 ESM (Boucher et al., 2020). In addition, through a set of crossed multi-factorial simulations, we also aim at estimating the CLCS dispersions associated specifically to climate and $[CO_2]$ (CCO2), land-use change (LUC) and nitrogen inputs (NIN) trajectories. We first present the ORCHIDEE-v3 model, the forcing datasets used as well as the modelling protocol and computed metrics used in the study (Section 2). We then present and discuss the CLCS resulting from our set of simulations and their sources of dispersion, globally and for eight large regions (Section 3). Last, some recommendations are drawn in the perspective of the next CMIP exercise (Section 4).

## 2 Methods

### 2.1 The ORCHIDEE-v3 model

ORCHIDEE (Organizing Carbon and Hydrology In Dynamic Ecosystems) is a global process-based terrestrial ecosystem model used to quantify energy, water, carbon and nitrogen flows and associated stocks in the soil-vegetation-atmosphere continuum (Krinner et al., 2005; Vuichard et al., 2019). For the last CMIP6 exercise (Boucher et al., 2020), ORCHIDEE-v2, a carbon-only version of ORCHIDEE has been used as the land component of the Earth System Model of the Institut Pierre Simon-Laplace (IPSL-CM6). ORCHIDEE-v3 is an advanced version in which N cycle and the C-N interactions have been included (Vuichard et al., 2019). ORCHIDEE-v3 needs as input data, information about climate (near-surface air temperature, precipitation, short and long-wave incoming radiation, specific air humidity), atmospheric $CO_2$ concentration, land cover, but also atmospheric N deposition (NHx and NOy) and N fertilizer rates on managed lands. ORCHIDEE-v3 showed good performance at simulating Gross Primary Productivity (GPP) and Leaf Area Index (LAI) both at site and global scales (Vuichard et al., 2019). It also ranked with a good score for a set of key land variables in a recent model benchmark study (Seiler et al., 2022) as well as in the TRENDY model inter-comparison project as part of the land surface models contributing to the Global Carbon Budget (Friedlingstein et al., 2022).

### 2.2 Model input datasets

Inputs related to atmospheric $CO_2$ concentration ($[CO_2]$), land-use, wood harvest, N-fertilizer and nitrogen deposition are those used for the historical and the different SSP CMIP6-related experiments and stored on input4MIPs nodes (https://esgf-node.llnl.gov/projects/input4mips/). Land-use, wood harvest and N-fertilizer input data are produced by the Land-Use Harmonization 2 (LUH2) project (Hurtt et al., 2020). Land use information from LUH2 consists of fractions of grid cell area at 0.25° resolution for cropland (five sub-categories), managed pasture, rangeland, urban, primary and secondary forested and non-forested land. The procedure needed for translating the original data for land-use into the fifteen land classes of

ORCHIDEE is described in Lurton et al. (2020). In this procedure, information regarding the cropland and pasture areas from LUH2 is preserved while natural land is split into the different unmanaged land classes of ORCHIDEE using data from the ESA CCI Land cover product for the year 2016 (ESA, 2022). For the period 2015-2100, LUH2 data is based on land use and land management information from the eight different marker scenarios generated by IAMs. Nitrogen deposition fields are produced by the CAM-Chem climate-chemistry model (Hegglin M. et al., 2016) with emission data from the different marker

scenarios for the period 2015-2100. Climate data used as inputs of the land-only ORCHIDEE-v3 simulations correspond to the IPSL-CM6A-LR model outputs (at a global resolution of 2.5°x1.27° in longitude and latitude) for the historical and the different SSP CMIP6 experiments. In this study, ORCHIDEE-v3 ran at the same resolution as the climate input data. Figure 1 summarises the modelling framework developed for this study with the different input data used.

## 2.3 Reference simulations

In order to get C and N vegetation and soil pools at equilibrium, we ran a spin-up simulation with the boundary conditions of year 1850 but recycling climate data for the period 1850-1869 in order to account for an inter-annual variability. From this equilibrium state, simulations ran for the historical period (1850-2014) and for each of the eight SSP experiments from 2015 to 2100.

## 2.4 Land-use and Nitrogen inputs related sensitivity simulations

The objective was to investigate the impact on CLCS of the uncertainty associated to land-use and nitrogen inputs (ie., atmospheric N deposition and N fertilization) for a given SSP $p$. Given that all gridded harmonized data for the land-use and nitrogen inputs of non-marker scenarios of SSP $p$ are not available, we used the gridded data (for land-use and nitrogen inputs) from marker scenarios of selected alternative SSPs to assess the sensitivity of the projected land carbon store for SSP $p$ to land-use and nitrogen inputs uncertainties. In other words, we used the selected SSP markers spread as a proxy for the inter-IAM

spread regarding the land-use and nitrogen inputs trajectories for any given SSP. This is a strong assumption but supported by the comparison between inter-SSP markers and inter-IAM trajectories for the different SSPs. The comparison has been conducted for the following variables : forested land area (Fig. 2), NHx emissions (Fig. 3), cropland area (Fig. A1), pasture land area (Fig. A2) and NOy emissions (Fig. A3). Given that, ultimately, we would like to assess the uncertainty associated to land-use and nitrogen inputs from the different IAMs for any SSP, in the following we may use the term "uncertainty" when

referring to the different inter-SSP markers trajectories although they correspond more to certain trajectories obtained for different assumptions in terms of socio-economic development and mitigation level.

Due to computing time resources, we limited our sensitivity study to four SSP markers among the eight available and for each of these four SSPs, we used the land-use and nitrogen inputs trajectories of this set of four SSP markers to assess their impacts on CLCS. The selected SSPs were SSP1-1.9, SSP3-7.0, SSP4-3.4 and SSP5-8.5. The selected SSP markers were computed by

the Integrated Assessment Models IMAGE, AIM/CGE, GCAM4 and REMIND-MAGPIE, respectively. IAMs are driven by projections of economic growth and population but differ in their representation of socio-economic, energy- and land-related

processes. Information on IAM modelling regarding land-use allocation and nitrogen emissions can be found in Tables A4 and A5, respectively. We selected these four SSPs because 1/ they encompass a large spread of $CO_2$ level in 2100 ranging from 394 to 1135 ppm; and 2/ the inter-IAM spreads for land-use but also N-related input data trajectories from this selection

are comparable to those from the eight SSPs.

Based on the IAM output data produced for CMIP6 available on the SSP Database (https://tntcat.iiasa.ac.at/SspD), we showed that the inter-selected SSP markers spread of the forested global land area in 2100 is narrower than the inter-IAM spread for six out of eight SSPs (Fig. 2). Similarly, the inter-selected SSP markers spread of the global NH3 emissions in 2100 is narrower than the inter-IAM spread for seven out of the eight SSPs (Fig. 3). However, for some variables simulated by IAMs, the inter-

selected SSP markers spread is significantly larger than the inter-IAM spread for many SSPs. This is particularly the case of NOy emissions (Fig. A3) for which the inter-selected markers spread is larger than the inter-IAM spread for any of the eight SSPs. Thus, depending of the driving variable considered (forested lands, pasture or croplands, NH3 or NOy emissions) and of the SSP considered, the use of the selected SSP markers spread as a proxy may translate into an upper or lower estimate of the inter-IAM spread. Overall, our assumption of using the land-use and N trajectories of the different SSP markers as a

surrogate of the trajectories simulated by the different IAMs for each SSP looks reasonable (from the above analysis). The comparison between inter-SSP markers and inter-IAM trajectories for the different SSPs is presented at global scale, but the conclusion that the selected SSP markers spread is comparable to the inter-IAM spread for the different SSPs remains valid at regional scale (based on the data available on the SSP Database for five aggregated regions ("Asia", "Latin America", "Reforming economies", "Middle East and Africa" and countries from the "Organisation for Economic Co-operation and

Development"), not shown).

In addition, using alternative SSP scenarios for a given driving variable (for instance LUC or nitrogen atmospheric deposition) while keeping the other driving variables from a single SSP may break down the coherency between driving variables, as established within each IAM. However, we showed that while $NH_3$ emissions show a good linear relationship with cropland area for most of the IAMs, the slope of this relationship is significantly different across IAMs (Fig. A6). This indicates that no

common and unique relationship exists across IAMs and thus using the marker SSP spread for each variable independently of the others is a reasonable assumption.

### 2.5 Metrics assessing the change in land carbon store and its sensitivity to different land-use and nitrogen inputs

We analysed specifically the projected change in land carbon store (CLCS) for the four selected pathways and its sensitivity

to the different land-use and N-inputs marker trajectories from these selected SSPs. To perform this analysis, we ran a set of sixteen sensitivity simulations for each of the four selected reference simulations, where land-use and N-related data from the four SSPs is used independently as forcing (four land-use trajectories times four N-inputs trajectories). The trajectories over

2015-2100 of input data for forested land area, total atmospheric nitrogen deposition, nitrogen fertilizer application, atmospheric $[CO_2]$ and near-surface temperature are shown on Fig. A7, A8, A9, A10 and A11, respectively.

We expressed CLCS as a function of climate and atmospheric $[CO_2]$ (CCO2), land-use change (LUC) and nitrogen inputs (NIN) trajectories (*CLCS(CCO2, LUC, NIN)*) and quantified the impact of CCO2, LUC and NIN trajectories on CLCS by computing mean ($\mu$) and standard deviation ($\sigma$) metrics based on the following equations:

$$X_{CLCS,CCO2}(j,k) = X\{CLCS(i,j,k)\}_{i=1-1.9,3-7.0,4-3.4,5-8.5}, \tag{1}$$

$$X_{CLCS,LUC}(i,k) = X\{CLCS(i,j,k)\}_{j=1-1.9,3-7.0,4-3.4,5-8.5}, \tag{2}$$

$$X_{CLCS,NIN}(i,j) = X\{CLCS(i,j,k)\}_{k=1-1.9,3-7.0,4-3.4,5-8.5}, \tag{3}$$

where X stands for $\mu$ or $\sigma$; and the indices $i$, $j$ and $k$ stand for CCO2, LUC and NIN trajectories, respectively, each spanning the different SSPs.

From the above generic equations, we can further quantify the mean CLCS and standard deviation associated specifically to different land-use (LUC) and different atmospheric N deposition and fertilisation (NIN) trajectories, for each of the four

selected SSPs ($s$), $X_{CLCS,LUC}^{s}$ and $X_{CLCS,NIN}^{s}$, defined as:

$$X_{CLCS,LUC}^{s} = X_{CLCS,LUC}(i=s,k=s) \tag{4}$$

$$\text{and } X_{CLCS,NIN}^{s} = X_{CLCS,NIN}(i=s,j=s) \tag{5}$$

for $s$ = 1-1.9, 3-7.0, 4-3.4 and 5-8.5.

We also quantified the CLCS and standard deviation associated to land-use plus atmospheric N deposition and fertilisation

(LUC+NIN), $X_{CLCS,LUC+NIN}^{s}$. It is written as:

$$X_{CLCS,LUC+NIN}^{s} = X\{CLCS(i=s,j,k)\}_{j,k=1-1.9,3-7.0,4-3.4,5-8.5}, \tag{6}$$

In order to report on the overall dispersion of CLCS and the contribution from the three drivers (CCO2, LUC and NIN), we first computed $\mu$ and $\sigma$ accounting for all drivers:

$$X_{CLCS,TOT} = X\{CLCS(i,j,k)\}_{i,j,k=1-1.9,3-7.0,4-3.4,5-8.5}, \tag{7}$$

We then computed the mean standard deviation, $\overline{\sigma}_{CLCS,D}$ in order to quantify the impact on CLCS of each of the three drivers (D being CCO2, LUC or NIN) irrespective of the combinations of the two others:

$$\overline{\sigma}_{CLCS,CCO2} = \mu\{\sigma_{CLCS,CCO2}(j,k)\}_{j,k=1-1.9,3-7.0,4-3.4,5-8.5}, \tag{8}$$

$$\overline{\sigma}_{CLCS,LUC} = \mu\{\sigma_{CLCS,LUC}(i,k)\}_{i,k=1-1.9,3-7.0,4-3.4,5-8.5}, \tag{9}$$

$$\overline{\sigma}_{CLCS,NIN} = \mu\{\sigma_{CLCS,NIN}(i,j)\}_{i,j=1-1.9,3-7.0,4-3.4,5-8.5}, \tag{10}$$

Last, we expressed the relative impact on the CLCS spread of each of the three drivers, $r_{CLCS,D}$ as:

$$r_{CLCS,D} = \frac{\overline{\sigma}_{CLCS,D}}{\overline{\sigma}_{CLCS,CCO2} + \overline{\sigma}_{CLCS,LUC} + \overline{\sigma}_{CLCS,NIN}} \times 100, \tag{11}$$

for D = CCO2, LUC and NIN

## 3 Results and discussion

**3.1 Change in land carbon store (CLCS) over the historical period and for the different SSPs experiments**

The change in land carbon store (CLCS) simulated by ORCHIDEE-v3 over the historical period (1850 – 2014) corresponds to a small loss of carbon in the land reservoir of 7.7 PgC (table 1 where a negative value corresponds to a source to the atmosphere). This results from a C source due to land-use change larger than the land C sink induced by the increasing [$CO_2$] and N deposition. Over the period 1850-2100 and depending of the SSP, the CLCS varies between a small source of 5.6 PgC

(SSP4-3.4) to a land sink of 115.5 PgC (SSP5-8.5). The CLCS simulated by ORCHIDEE-v3 are in the low-end range of the values reported by Liddicoat et al. (2021) with an ensemble of nine ESMs (table 1 to be compared to table S3 of Liddicoat et al., 2021). ORCIHDEE-v3's CLCS is very similar to the one simulated by UKESM1-0-LL for the historical period and for any of the seven SSPs studied by this ESM. The CLCS standard deviation induced by considering different N-related trajectories is relatively similar irrespective of the SSP considered with $\sigma^s_{CLCS,NIN}$ values for the period 1850-2100 varying

between 10.9 and 13.6 PgC depending on the SSP (Table 1 and Fig. 4). The effect of considering different LUC-related trajectories on the CLCS is more important with a standard deviation ($\sigma^s_{CLCS,LUC}$ for 1850-2100) going from 38.1 PgC (for SSP1-1.9) to 46.2 PgC (for SSP5-8.5). Accounting for both sources of uncertainty (LUC and NIN) on CLCS leads to a similar dispersion than considering LUC uncertainty only with $\sigma^s_{CLCS,LUC+NIN}$ varying between 37.2 and 45.3 PgC depending on the SSP (Table 1). Expressed as a percentage of the mean CLCS from 2015 to 2100, these values correspond to standard deviations

ranging between 43.8% (for SSP5-8.5) and 114.1% (for SSP1-1.9) of $\mu^s_{CLCS,LUC+NIN}$. For SSP1-1.9 with a relative dispersion

higher than 100%, accounting for the spread on LUC and NIN has the capacity of turning CLCS from a gain to a loss of carbon. Although important, these CLCS dispersions induced by the LUC and NIN trajectories are a factor 2 to 3 less than those associated to the multi ESM ensemble assessed by Liddicoat et al. (2019) for all four studied SSPs except SSP1-1.9. Based on the data reported by Liddicoat et al. (2019, table S3), the CLCS standard deviation of the multi ESM ensemble over the period 2015-2100 equals to 39.6 PgC (52% of the multi ESM ensemble mean), 123.5 PgC (63%), 86.9 PgC (381%) and 162.3 PgC (58%) for SSP1-1.9, SSP3-7.0, SSP4-3.4 and SSP5-8.5 respectively).

As shown on Fig. 4 (right-side plot of each panel), depending of the LUC and NIN trajectories associated to the marker scenarios, the CLCS from 2015 to 2100 estimated for the marker may be in the very low-end range of values for all NIN and LUC combinations (SSP4-3.4), in the high-end range (SSP1-1.9) or closed to the mean value $\mu^s_{CLCS,LUC+NIN}$ (SSP3-7.0 and to some extent SSP5-8.5).

**3.2 Spatial and temporal analysis of the CLCS dispersion and its drivers**

When accounting for all combinations of NIN, LUC and CCO2 trajectories, the global CLCS at the end of the 21$^{st}$ century ranges from a source of 33 PgC to a sink of 179 PgC (Fig. 5, envelope of the white transparent areas with right y-axis). The mean change by 2100 (relatively to 2014) in carbon stored in land ($\mu_{CLCS,TOT}$) as well as its standard deviation induced by the different driver trajectories ($\sigma_{CLCS,TOT}$) vary significantly spatially with a large contribution from Africa and Tropical Asia (and especially tropical forests) to both the mean and standard deviation (Fig. 6). The CLCS spread induced by the different LUC trajectories ($\sigma_{CLCS,LUC}$) is slightly larger than the one related to the CCO2 trajectory ($\sigma_{CLCS,CCO2}$). On average for all combinations of NIN, LUC and CCO2, the relative impact of LUC on the CLCS spread ($r_{CLCS,LUC}$) amounts to 48% globally at the end of the 21$^{st}$ century, while $r_{CLCS,CCO2}$ value is about 38% (Fig. 5, coloured areas with left y-axis). The relative impact of NIN on the CLCS spread is one third less, with a value of $r_{CLCS,NIN}$ equals to 14%. The relative impacts of the three drivers on the CLCS spread at the end of the 21$^{st}$ century show contrasted results at regional scale (Temporal evolution for the eight global regions in Fig. 5 and spatial distribution in 2100 in Fig. 7). In Africa and Tropical Asia regions, where the strength of the land use change varies significantly from one SSP to another, the relative impact of LUC is far more important than the impact of CCO2 (and NIN) with values of $r_{CLCS,LUC}$ of ~74% for both regions (Fig. 5 and 7). As a consequence, the value of

$r_{CLCS,CCO2}$ in these two regions is less than 20% by 2100. They are the only two regions for which CLCS shifts significantly from source to sink depending of the LUC trajectories (Fig. 5, envelope of the white semi-transparent area) with regional $\mu_{CLCS,TOT} \pm \sigma_{CLCS,TOT}$ values of -18±27 PgC and 5±14 PgC by 2100, for Africa and Tropical Asia region respectively. Due to the strong impact of LUC on CLCS (Fig. 5 and 7) and its large area (Fig. A12), Africa is the region that contributes the most to the overall dispersion of CLCS globally ($\sigma_{CLCS,TOT}$ of 27 PgC for Africa, to be compared to $\sigma_{CLCS,TOT}$ of 53 PgC for the

globe). For the six other regions where the impact of LUC is less important, CCO2 is the factor that drives the most the CLCS dispersion with $r_{CLCS,CCO2}$ values ranging from 37% (for Europe) to ~57.5% (for "Boreal Asia" and "Australia and New Zealand" regions). In these regions, the impact of NIN on the CLCS dispersion varies significantly depending on how the atmospheric N deposition trajectories are contrasted within a region but also on how the terrestrial ecosystems are N-limited regionally. In "South America" and "Australia and New Zealand" regions, the relative impact of NIN is very small with

$r_{CLCS,NIN}$ values less than 10%. In the other four regions, $r_{CLCS,NIN}$ values are larger than 23% and up to 35% for the "Boreal Asia" region.

The time evolution of the relative impacts of the three drivers on the CLCS dispersion is not uniform over the 21[st] century (Fig. 5). Globally, $r_{CLCS,CCO2}$ decreases over the two first decades (2015-2030, from values greater than 50% down to 7%) and increases the following decades with a kind of Michaelis-Menten curve shape. Mirroring the time evolution of the relative

impact of CCO2, $r_{CLCS,NIN}$ and $r_{CLCS,LUC}$ increase over the first decades of the 21[st] century and decrease after 2030 and 2040 for NIN and LUC respectively. These specific temporal dynamics, which result from the combination of specific time evolution and time-response on the CLCS of the three studied drivers, are obtained globally but also for most large regions (eg Temperate Asia, North America, South America). These first-decades dynamics are not analysed in more details here as they correspond to periods over which the CLCS overall dispersion remains small (see time evolution of $\mu_{CLCS,TOT} \pm \sigma_{CLCS,TOT}$, envelope of the

white semi-transparent area on Fig. 5).

### 3.3 Change in carbon stored in vegetation and litter and soil pools

Further analysis showed that vegetation (above- and below-ground) is the reservoir contributing the most to CLCS (compared to soil and litter carbon reservoirs, Fig. 6). On average for all combinations of NIN, LUC and CCO2, the global change in

vegetation carbon store (CVCS) amounts to 47 PgC at the end of the 21$^{st}$ century (Fig. A13, middle of the white semi-transparent envelope), while the change in soil and litter carbon store (CSCS) amounts to 21 PgC (Fig. A15). The overall dispersion of CVCS globally is also much larger than the one of CSCS in 2100 ($\sigma_{CVCS,TOT}$ of 52 PgC, to be compared to $\sigma_{CSCS,TOT}$ of 9 PgC, see Fig. A13 and A15 and Fig. 6). Thus, vegetation is also the reservoir which contributes the most to the overall dispersion of CLCS ($\sigma_{CLCS,TOT}$ of 53 PgC for the globe). Carbon in vegetation being mostly stored in trees, forested lands are the main location of CVCS, while grasslands and croplands have only a marginal contribution to CVCS (Fig. A14).

On average, the relative impacts of CCO2, LUC and NIN on the CVCS spread are comparable to those on the CLCS spread with values for $r_{CVCS,CCO2}$, $r_{CVCS,LUC}$ and $r_{CVCS,NIN}$ equal to 45%, 48% and 7% respectively (Fig. A13). Note however that the relative impact of NIN on the CVCS spread is significantly lower than the one on the CLCS spread, globally ($r_{CVCS,NIN}$ of 7%, to be compared to $r_{CLCS,NIN}$ of 14% for the globe) but also regionally (for instance in the "Europe" or "Boreal Asia" regions). Compared to the results obtained for the CLCS and CVCS, the relative impacts of CCO2, LUC and NIN on the CSCS spread are very different (Fig. A15 and 7). NIN is the driver inducing the largest dispersion of CSCS globally ($r_{CSCS,NIN}$ of 41%) and in several regions (Europe, Boreal Asia, Temperate Asia and North America, see Fig. A15). The relative impacts of CCO2 and LUC on the global CSCS dispersion share equally the remaining percentages with values of 29% and 30% for $r_{CSCS,CCO2}$ and $r_{CSCS,LUC}$, respectively (Fig. A15). The lower relative impact of LUC on the CSCS dispersion compared to the CVCS dispersion can be explained by the fact that land-use changes impact more significantly the standing biomass than the modelled soil organic carbon dynamic. For the effect of CCO2, a deeper analysis (not shown) revealed that CCO2 is driving the soil carbon store via two opposite contributions. Soil carbon store increases with atmospheric [$CO_2$] increase while it decreases with soil temperature increase due to higher soil organic decomposition rate. The compensating effects of atmospheric [$CO_2$] and soil temperature result in limited changes in soil carbon store for the different CCO2 scenarios, in which soil temperature varies proportionally to atmospheric [$CO_2$].

### 3.4 CLCS as a function of atmospheric CO₂, forested land area and atmospheric nitrogen deposition

The ensemble of sixty-four factorial simulations offers the advantage to isolate and quantify the effect of one specific driver among the three considered in this study (CCO2, LUC and NIN) which are otherwise mixed up in the standard reference SSP

simulations. We express CLCS in 2100 (i.e., the total change from 2015 to 2100) as a function of one driver (atmospheric [$CO_2$] for CCO2, forested lands for LUC or N atmospheric deposition for NIN, in 2100) for the sixteen simulations driven by the different combinations of the two other drivers (Fig. 8). The different relationships between CLCS and any of the three drivers are similar irrespective of the simulations considered meaning that there is no strong co-varying effects across drivers. Only the CLCS baseline level differs between simulations. The CLCS response curve to [$CO_2$] shows a saturation effect for the highest $CO_2$ level (~1100 ppm) driven by the limitation of C assimilated by photosynthesis at high [$CO_2$]. Based on a simple linear regression, the CLCS response to $CO_2$ equals 0.1 PgC ppm$^{-1}$ (Fig. 8a). Note that this sensitivity cannot be compared to the well-studied land carbon–concentration feedback metric ($\beta_L$, PgC ppm$^{-1}$) (Arora et al., 2020; Friedlingstein, 2015) since in our study the CLCS response to $CO_2$ includes also the indirect effect of [$CO_2$] on land carbon store via climate change and in particular temperature change.

We also highlight a relationship between the forested land area in 2100 and CLCS in 2100 (Fig. 8b). The forested land area in 2100 is inversely proportional to the deforestation trend (or proportional to the re/afforestation trend) experienced over the 21$^{st}$ century in the different SSPs. As a consequence, the higher forested land area, the higher CLCS. The relationship between CLCS and the forested land area is not strictly linear due to the different regions where the deforestation (or re/afforestation) acts in the SSPs, with different ecosystem productivity and vegetation carbon storage (higher storage for tropical ecosystems). However, on average, based on a linear regression, the CLCS response to the forested lands equals 13.85 PgC (Mkm$^2$ of forested lands)$^{-1}$ (Fig. 8b). Last, CLCS shows a nearly linear relationship with the global mean atmospheric N deposition rate in 2100 (Fig. 8c). The 2100 rate is used here as an indicator of the load of atmospheric N deposited on land over the 21$^{st}$ century and its fertilising effect on terrestrial ecosystems. This results in a CLCS response to N deposition of 1 PgC (TgN yr$^{-1}$)$^{-1}$.

### 3.5 Comparison with other studies and path for future research

To our knowledge, little attention has been paid to the co-effects of atmospheric [$CO_2$], atmospheric nitrogen deposition and land-use change on the change in land carbon store in the CMIP6 framework and how these drivers interplay together at global and regional scales. A 1pctCO2 experiment was part of the DECK ensemble (Eyring et al., 2016) in order to analyse the effects of a 1% yr$^{-1}$ increase in atmospheric [$CO_2$] on the radiative (RAD) and carbon cycle (BGC) components with pre-industrial atmospheric N deposition. In addition to the 1pctCO2 experiment, two experiments (namely 1pctCO2Ndep and 1pctCO2Ndep-

bgc) were planned in the Coupled Climate–Carbon Cycle Model Intercomparison Project (Jones et al., 2016) with time-increasing atmospheric N deposition, with the objective of quantifying the co-effects of atmospheric $CO_2$ and N deposition increases. Unfortunately, only three modelling groups performed these two additional experiments and no study made use of them so far. In the Land Use Model Intercomparison Project (Lawrence et al., 2016), the two experiments ssp370-ssp126Lu and ssp126-ssp370Lu, based on the ScenarioMIP ssp370 and ssp126 experiments but swapping their land-use datasets (Hurtt et al., 2020), aim at quantifying the specific contribution from land-use change on the climate and carbon cycle over the 21[st] century. With this set of 2x2 experiments, Ito et al. (2020) quantified the impact of land-use change on the total soil carbon stock (cSoil) simulated by seven ESMs. Although limited to only two contrasted land-use trajectories, they reported large intermodel spread with change on cSoil in 2100 between pair experiments (which differ only by their land-use trajectories) varying between -14 and +28 PgC depending on the ESM. The large inter-model spread regarding changes in land carbon store has also been reported in many studies such as the one of Liddicoat et al. (2021) based on the CMIP6 historical and SSPs experiments or the one of O'Sullivan et al. (O'Sullivan et al., 2022) based on the TRENDY land models ensemble over the last six decades. In this latter study, eighteen land surface models were used to assess the changes in carbon stored in vegetation and soil due to change in CO2 and Nitrogen deposition, climate, and land use. ORCHIDEE-v3 was one of these models and showed results very similar to those obtained with the multi model ensemble means which gives confidence on how relevant are the results of the present study. Nevertheless, there is a need of performing the multi-sensitivity analysis we proposed in this paper with an extended ensemble of models, in order to evaluate the robustness of our conclusions with other models that have different representations of the key C-related ecosystem processes.

## 4 Summary and conclusions

Our study aimed to quantify the impacts of the land-use- and nitrogen inputs-related IAM uncertainties on the change in land carbon store as simulated by the land component of an ESM, forced by climate projections. In the absence of harmonized and downscaled gridded information for the IAMs other than the marker one of each SSP, we used the land-use and nitrogen trajectories of the different SSP markers as a surrogate of the trajectories simulated by the different IAMs for each SSP. We showed that the spread of the simulated change in global land carbon store induced by the different land-use trajectories across

SSPs is slightly larger than the one associated with the different atmospheric [$CO_2$] trajectories. Globally, uncertainty associated with N inputs (mostly N depositions which originate from the N emissions) is responsible for a spread in the change in land carbon store that is lower by a factor three, than the one driven by atmospheric [$CO_2$] or land-use changes. The relative impact of these different uncertainties showed contrasted responses regionally. In regions with very contrasted land-use trajectories across SSPs, such as Africa, the spread in the change in land carbon store is mainly driven by land-use change. In contrast, in regions where land-use trajectories are more similar across SSPs, the impact of the nitrogen deposition-related uncertainty on the change in land carbon store may be almost as large as the one induced by uncertainty on atmospheric $CO_2$ and land-use changes. In addition, we separated the change in land carbon store between a change in the vegetation reservoir and a change in the soil plus litter C reservoirs, indicating a much larger contribution from the vegetation. Although we showed that the inter-marker spread and the inter-IAM spread for a given SSP were of the same order for the land-use trajectories but also for the N emissions trajectories globally, the two spreads are not strictly similar for each diagnostic variable by the IAMs or for each SSP. In this respect, there is a need for delivering harmonized and downscaled information about land-use changes, N emissions and N atmospheric deposition trajectories simulated by all IAMs for each SSP and not only by the marker IAMs. Performing sensitivity ESM or land-only experiments with these extra datasets is the only way to accurately assess the specific IAM-related uncertainty impacts on the carbon cycle and the climate system. While many GHG mitigation strategies rely more and more on land-based solutions, this calls for facilitating the communication and evaluation between IAM and ESM modelling frameworks. Making available additional IAM scenarios to be used in the next CMIP exercise should contribute to this objective. In addition, given the large impact of land use change differences between IAMs (for a given SSP) and the significant impact (although lower) of N inputs, we also recommend that the IAM community provides more information on the uncertainties associated to these drivers. For instance, it would be informative to obtain quantitative information on the uncertainty associated to these variables, with a high and a low range trajectory for each driver and whether these uncertainties stand from structural or parametric IAM uncertainties. Information on the level of correlation between the uncertainty associated to each driver (land use and N inputs) would also help to propagate them in the state variables of LSMs and ESMs simulations.

**Appendix A**

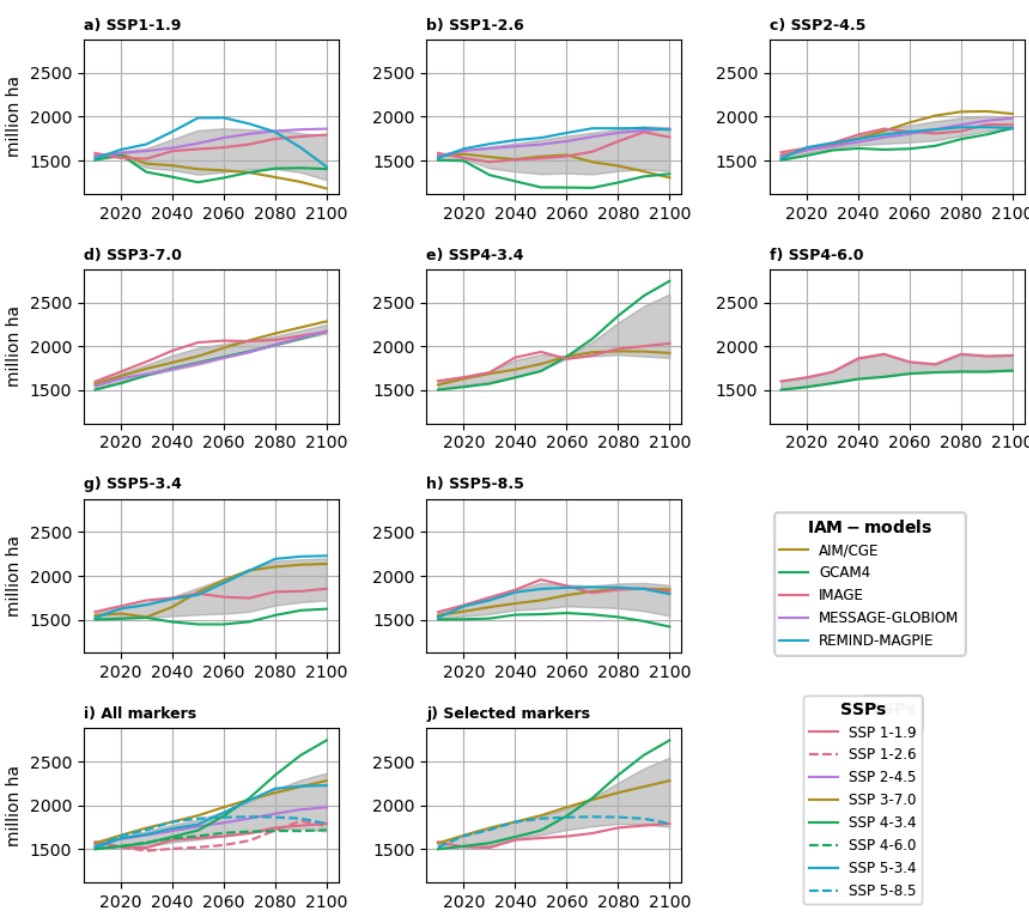

**Global cropland area**

**Figure A1: Time evolution (2015-2100) of the global cropland area (Mha) projected by (a to h) different Integrated Assessment Models (IAM) for different Shared Socio-economic Pathways, (i) all IAM markers and (j) the selected IAM markers used in the study. Grey aeras represent the time evolution of the mean ± sigma. Data from https://tntcat.iiasa.ac.at/SspDb**


**Global pasture land area**

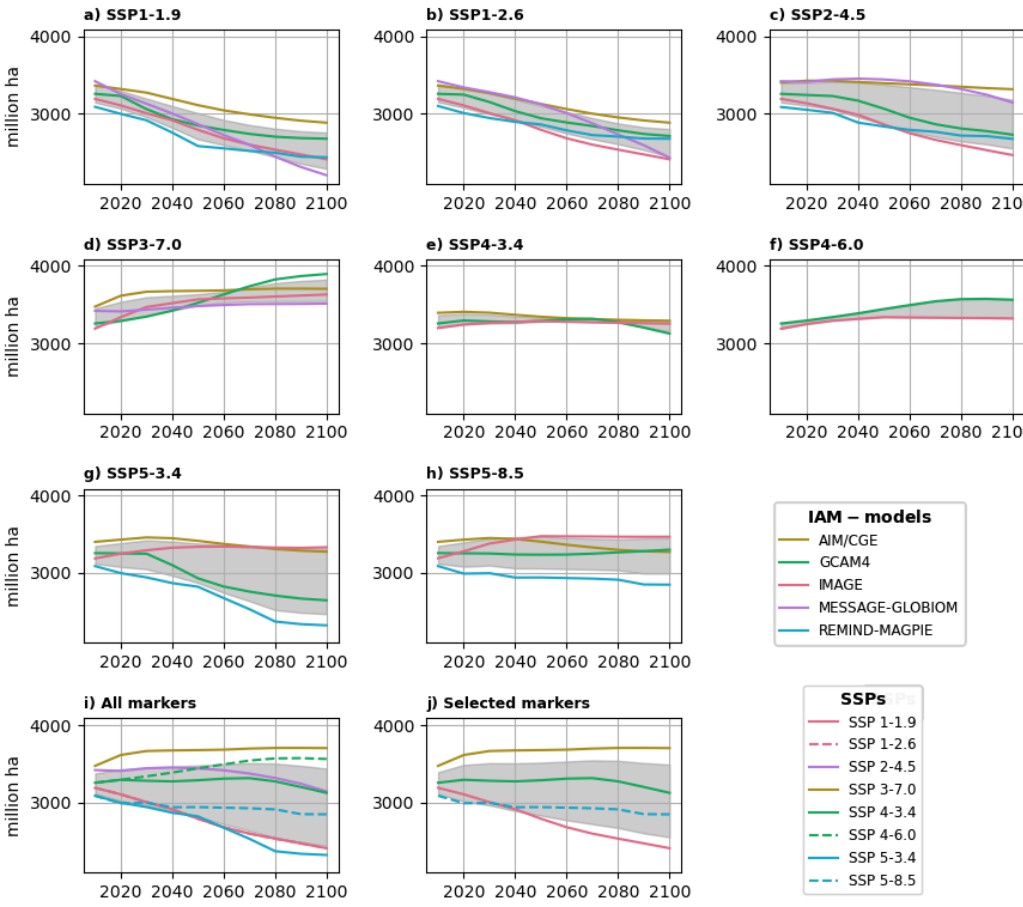

**Figure A2: Time evolution (2015-2100) of the global pasture land area (Mha) projected by (a to h) different Integrated Assessment Models (IAM) for different Shared Socio-economic Pathways, (i) all IAM markers and (j) the selected IAM markers used in the study. Grey aeras represent the time evolution of the mean ± sigma. Data from https://tntcat.iiasa.ac.at/SspDb**


**Global NOy emissions**

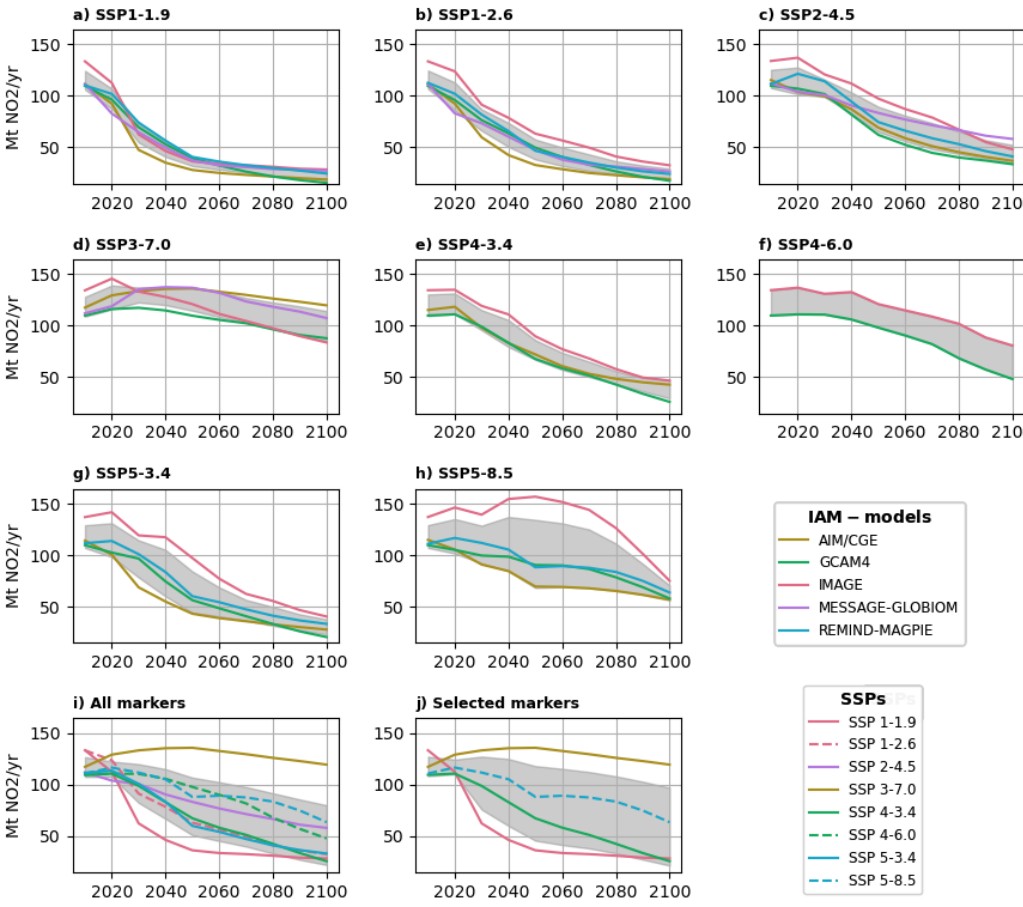

**Figure A3: Time evolution (2015-2100) of the global NOy (NO2) emissions (Mt(NO2) yr⁻¹) projected by (a to h) different Integrated Assessment Models (IAM) for different Shared Socio-economic Pathways, (i) all IAM markers and (j) the selected IAM markers used in the study. Grey aeras represent the time evolution of the mean ± sigma. Data from https://tntcat.iiasa.ac.at/SspDb**


| IAM | | IMAGE | AIM/CGE | GCAM4 | REMIND-MAGPIE |
|---|---|---|---|---|---|
| Used for marker SSP | | 1-1.9 | 3-7.0 | 4-3.4 | 5-8.5 |
| IAM category | | Recursive dynamic partial equilibrium model | Recursive dynamic general equilibrium model | Recursive dynamic partial equilibrium model | Intertemporal optimization general equilibrium model |
| Reference article | | van Vuuren et al., 2017 | Fujimori et al., 2017 | Calvin et al., 2016 | Kriegler et al., 2017 |
| Number of world regions | for economy and energy | 26 | 17 | 32 | 11 |
| | for agriculture | 30'x30' | | 283 | 10 |
| Land-use allocation description (Popp et al., 2017) | | Demand for bio-energy crops and other agricultural products are combined within each region to determine future land use. Land use is allocated at the grid level based on the spatially explicit attainable yields, and other suitability considerations. Attainable yields are computed by the LPJml model as a function of land and climate conditions and changes in technology. | Allocation of land by sector is formulated as a multi-nominal logit function (Fujimori *et al.* 2014) to reflect differences in substitutability across land categories with land rent. | Land is allocated based on profit maximization with an assumption of non-linear distributions of profits for each competing use. Demand for bioenergy is determined by the energy system component of GCAM, which is fully integrated with the agriculture and land use component. GCAM allows for global trade in crops, forestry, and bioenergy. | The objective function of MAgPIE (Model of Agricultural Production and its Impacts on the Environment) is the fulfilment of agricultural demand for each region at minimum global costs under consideration of biophysical and socio-economic constraints. For meeting the demand, MAgPIE endogenously decides, based on cost-effectiveness, about intensification of agricultural production, cropland expansion and production relocation (intra-regionally and inter-regionally through international trade) |

**Table A4: General and land-use related information for the four Integrated Assessment Models specifically used in this study (adapted from Popp et al., 2017 and Rao et al., 2017)**

| Source | IMAGE | | | GCAM4 | | | AIM/CGE | | | REMIND-MAGPIE | | |
|---|---|---|---|---|---|---|---|---|---|---|---|---|
| | Activity | $NO_x$ | $NH_3$ | Activity | $NO_x$ | $NH_3$ | Activity | $NO_x$ | $NH_3$ | Activity | $NO_x$ | $NH_3$ |
| **Energy related** | | | | | | | | | | | | |
| End-use energy use (industry, transport, residential, services and other) | Energy consumption | EF | | Energy consumption | EF | EF | Energy consumption | EF | | Energy consumption | EF | EF |
| Energy sector (production of power, hydrogen, coal, oil, gas, bioenergy) | Energy production | EF | | Energy production | EF | EF | Energy production | EF | | Energy production | EF | EF |
| Other energy conversion | Energy conversion | EF | | Energy conversion | EF | EF | Energy conversion | EF | | Energy conversion | EF | EF |
| **Industry Non combustion** | | | | | | | | | | | | |
| Emissions from industrial process | Industry value added (IVA) | EF | | Industry value added (IVA) | EF | EF | Industry output | EF | | Exog. (GAINS)Industry value | EX | EX |
| Cement and Steel | Regional production | | | Regional production | EF | EF | Regional production | EF | | Exog. (GAINS)Industry value | EX | EX |
| **Agriculture and land-use related** | | | | | | | | | | | | |
| Animal waste, all animal categories | Number of animals | GE | GE | Production of live stock products | | EF | Production of live stock products | EF | EF | Nr. of animals, feed, exog. assumptions on changes in animal waste management | EF | EF |
| Landfills | | | | | | | | | | Population, GDP, exogenous | EX | |
| Deforestation | Carbon burnt | GE | GE | Size of forest OR change in size of forest | EF | EF | Forestry otput | EF | EF | Land-use change | GE | GE |
| Agricultural waste burning | Carbon burnt | GE | GE | Agricultural production | EF | EF | agricultural output | EF | EF | Crop residues burnt | GE | GE |
| Traditional biomass burning | Carbon burnt | GE | GE | Traditional biomass consumption | EF | EF | agricultural output | EF | EF | Carbon burnt | EF | EF |
| Savannah burning | Carbon burnt | GE | GE | Grassland area | EF | EF | Carbon burnt | EF | EF | Pasture area | EF | EF |
| Domestic sewage treatment | | | | | | | | | | | EX | |
| Crops | Fertilizser and manure inputs | GM | GM | Crop production | EF | EF | crop production | EF | EF | Fertilizer , manure, other nitrogen inputs | EF | EF |
| Managed grassland | N fertiliser and manure input, crop type | GM | GM | | | | pasture land | EF | EF | N manure input | EF | EF |
| Indirect emissions | | | | | | | | | | N crops, fertiliser, manure input and animal waste management | EF | EF |
| **Other activities** | | | | | | | | | | | | |
| International Shipping | | EF | | | EF | | Energy consumption | EF | | | EX | EX |
| International Aviation | | EF | | | | | | | | | EX | EX |

**Table A5: Information on $NO_x$ and $NH_3$ emission modelling for the four Integrated Assessment Models specifically used in this study, with details about the categories and subcategories emitting $NO_x$/$NH_3$, the modelling approach used (EF, GE, GM or EX) and the activity data used (adapted from Rao et al., 2017). EF, GE, GM and EX stand for "Regional emission factor applied to the specified activity level", "Grid-specific emission calculated from gridded activity level and (regional) emission factor", "Gridded, model-based emission (statistical or process-based model)" and "Exogenous trajectory developed and implemented in model", respectively**


**NH3 emissions vs . Global cropland area**

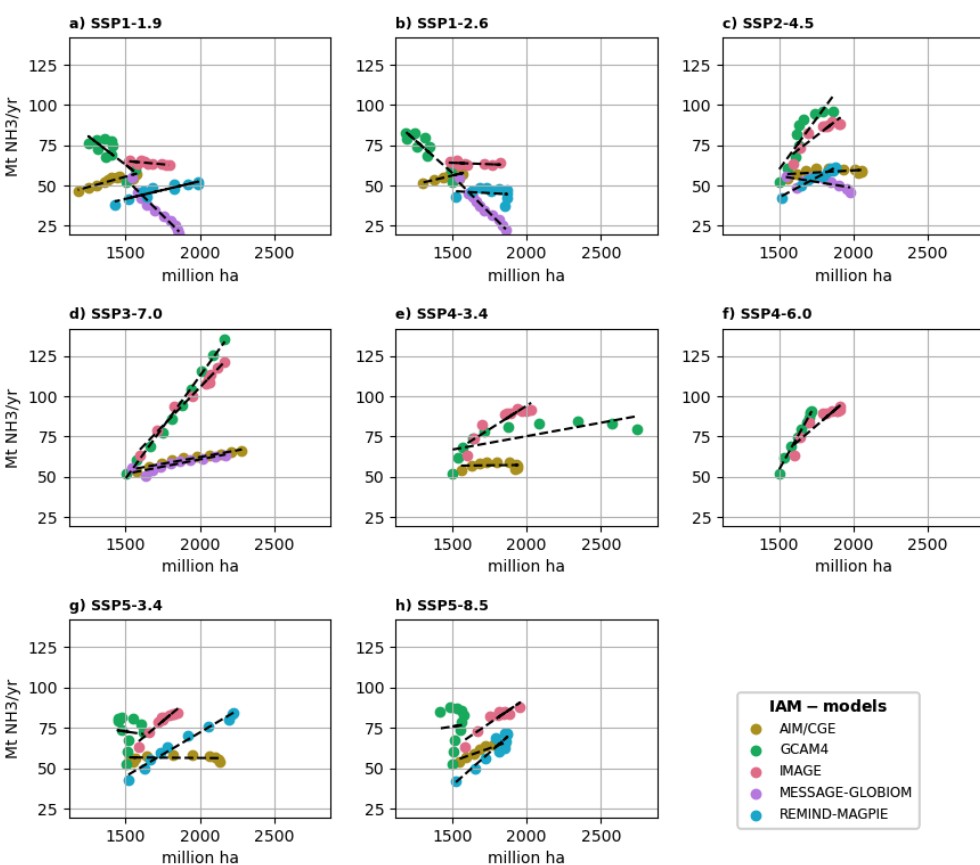

**Figure A6: NH₃ emissions (Mt(NH3) yr⁻¹)  as a function of global cropland area (millions of ha) projected by different Integrated Assessment Models (IAM) for different Shared Socio-economic Pathways. Data from https://tntcat.iiasa.ac.at/SspDb**


**Forested land area from 2015 to 2100**

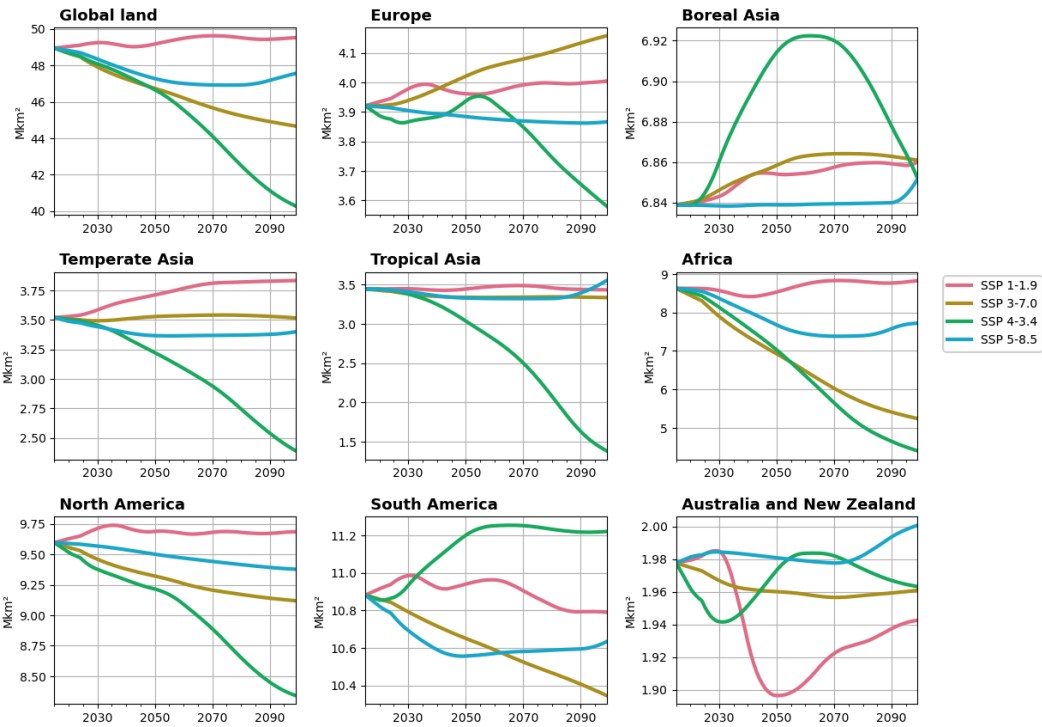

**Figure A7: Forested land area projected for SSP 1-1.9, 3-7.0, 4-3.4 and 5-8.5 over 2015-2100 and used as forcing of the ORCHIDEE-v3 model used in this study. Data from LUH2 project (Hurtt et al., 2020)**

**Total (NHx+NOy) deposition  from 2015 to 2100**

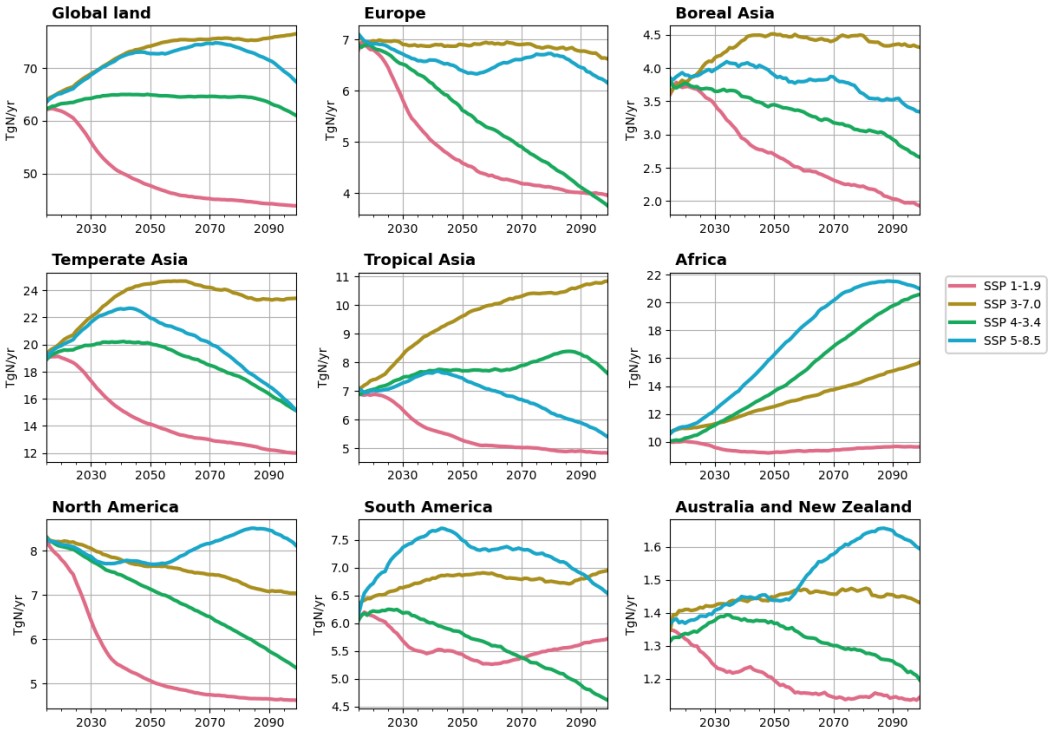

**Figure A8: Total atmospheric nitrogen deposition projected for SSP 1-1.9, 3-7.0, 4-3.4 and 5-8.5 over 2015-2100 and used as forcing of the ORCHIDEE-v3 model used in this study. Data from Hegglin et al. (2016)**


**Total nitrogen fertilizer application over croplands from 2015 to 2100**

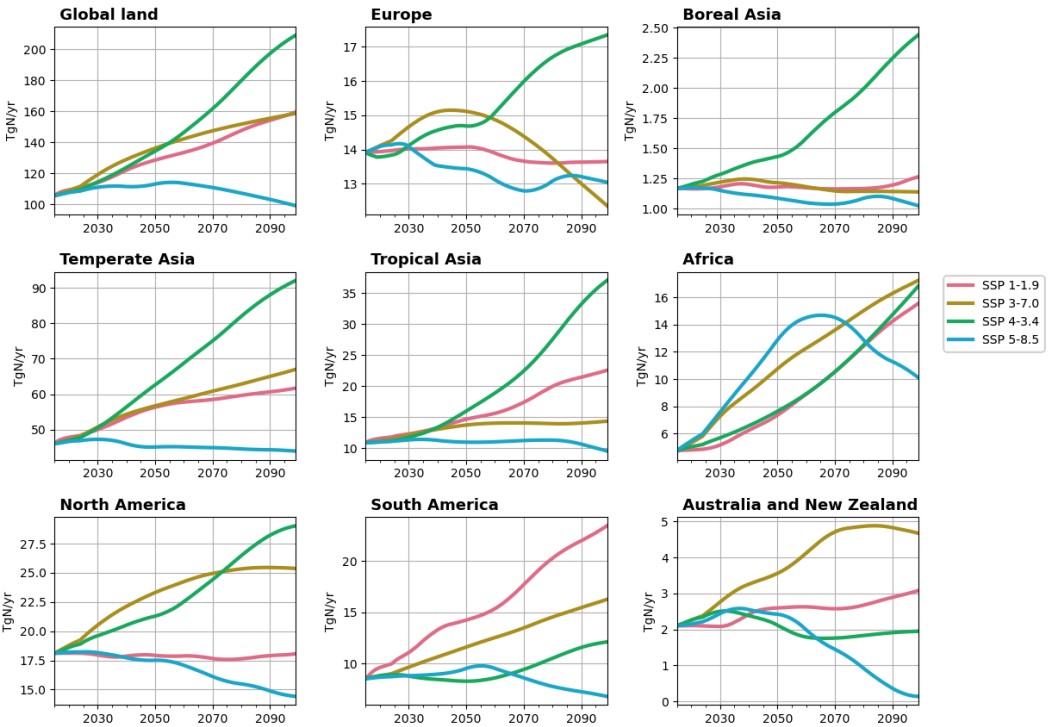

**Figure A9: Nitrogen fertilizer application projected for SSP 1-1.9, 3-7.0, 4-3.4 and 5-8.5 over 2015-2100 and used as forcing of the ORCHIDEE-v3 model used in this study. Data from LUH2 project (Hurtt et al., 2020)**


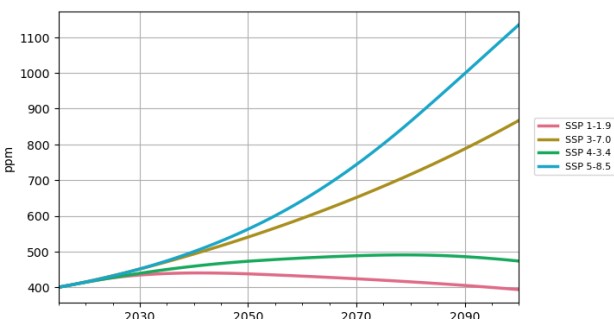

**Figure A10: Atmospheric CO₂ concentrations projected for SSP 1-1.9, 3-7.0, 4-3.4 and 5-8.5 over 2015-2100 and used as forcing of the ORCHIDEE-v3 model used in this study**

**Near-surface temperature over lands from 2015 to 2100**

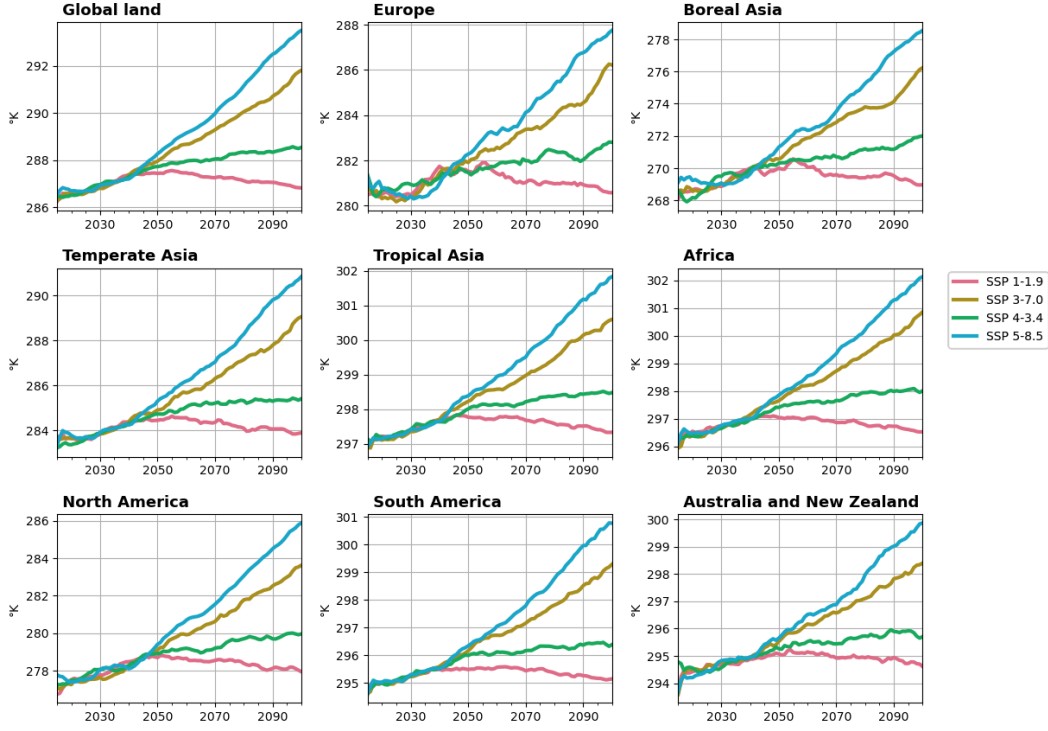


**Figure A11: Near-surface temperature projected by the IPSL-CM6 Earth System Model for SSP 1-1.9, 3-7.0, 4-3.4 and 5-8.5 over 2015-2100 and used as forcing of the ORCHIDEE-v3 model used in this study. Data from IPSL-CM6 (Boucher et al., 2020)**

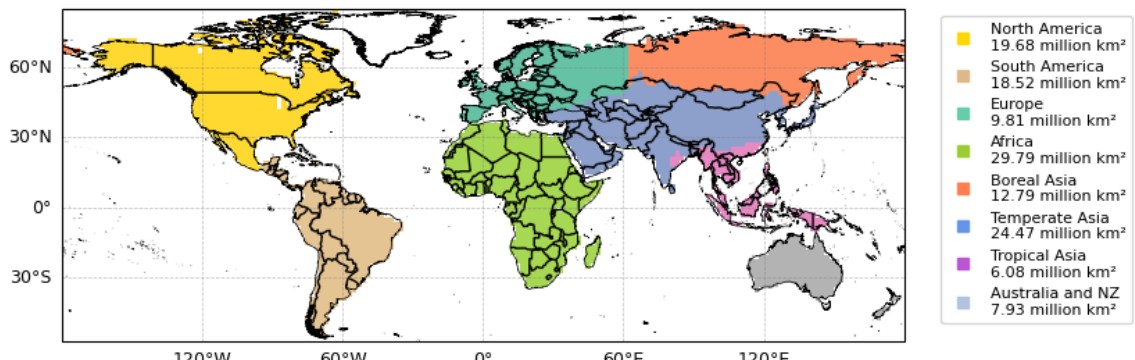

**Figure A12: Spatial distribution and size area of the eight regions used in the study.**

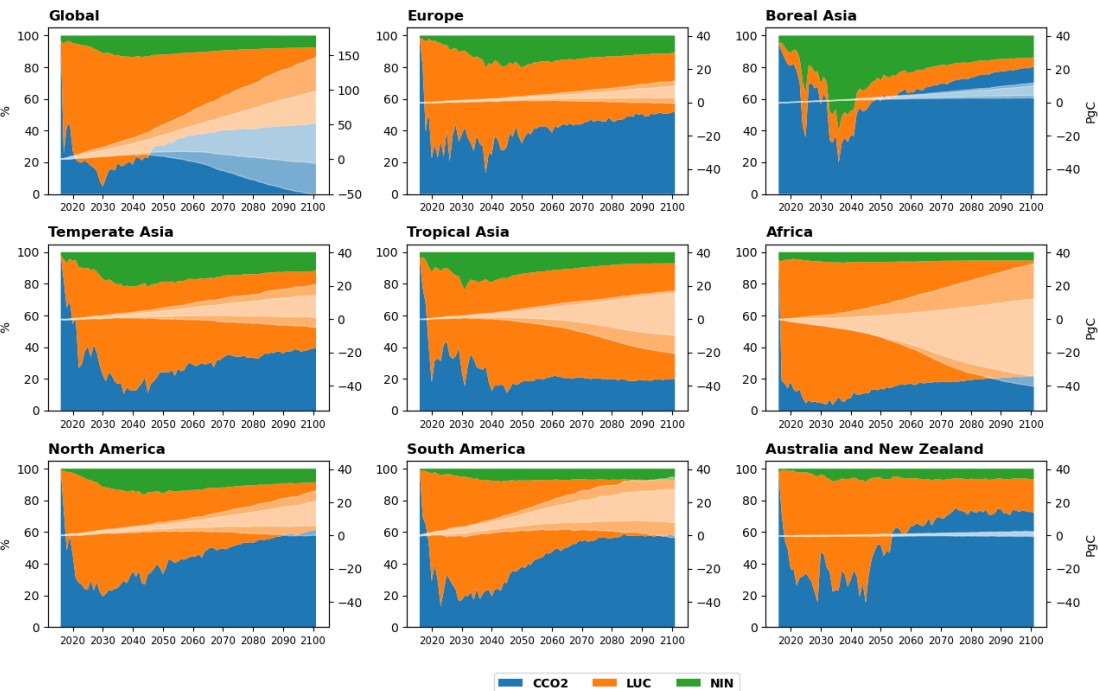

**Figure A13: Time evolution (2015-2100) of the change in vegetation carbon store (CVCS) accounting for different atmospheric [CO₂] and associated climate (CCO2), land-use change (LUC) and atmospheric N deposition and fertilisation (NIN) trajectories (with the white semi-transparent area representing $\mu_{CVCS,TOT} \pm \sigma_{CVCS,TOT}$ and the white transparent area representing the [min;max] of the ensemble of CVCS trajectories, in PgC, right y-axis) and the relative impact on the CVCS dispersion of the three drivers ($r_{CVCS,D}$, in percentage, left y-axis, with D being CCO2 (in blue), LUC (in orange) or NIN (in green)).**


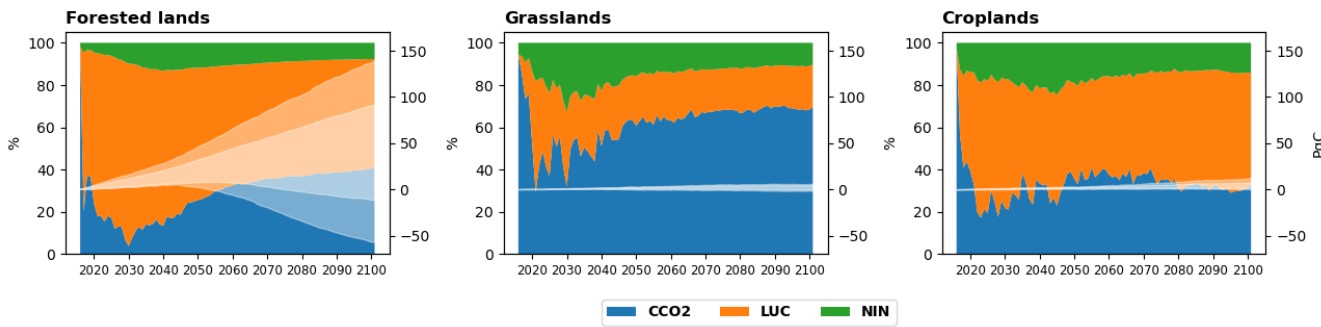

**Figure A14: Time evolution (2015-2100) of the global change in vegetation carbon store (CVCS) for tree, grass and crop cover accounting for different atmospheric [CO₂] and associated climate (CCO2), land-use change (LUC) and atmospheric N deposition and fertilisation (NIN) trajectories (with the white semi-transparent area representing $\mu_{CVCS,TOT} \pm \sigma_{CVCS,TOT}$ and the white transparent area representing the [min;max] of the ensemble of CVCS trajectories, in PgC, right y-axis) and the relative impact on the CVCS dispersion of the three drivers ($r_{CVCS,D}$, in percentage, left y-axis, with D being CCO2 (in blue), LUC (in orange) or NIN (in green)).**


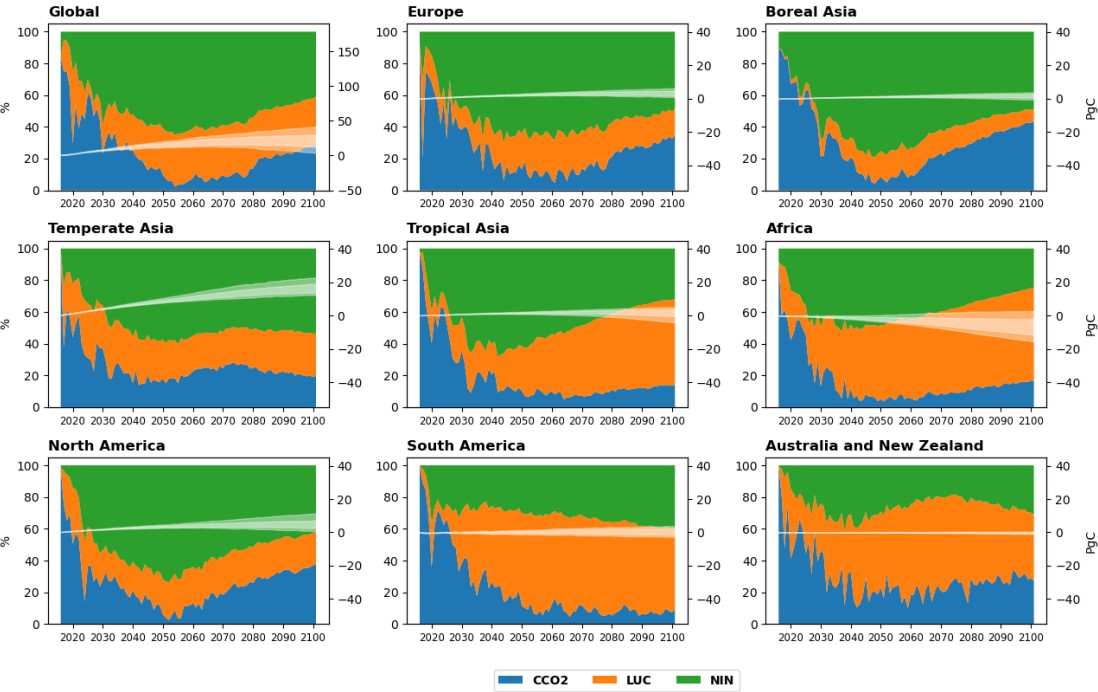

**Figure A15: Time evolution (2015-2100) of the change in litter and soil carbon store (CSCS) accounting for uncertainty on atmospheric [CO$_2$] and associated climate (CCO2), land-use change (LUC) and atmospheric N deposition and fertilisation (NIN) trajectories (with the white semi-transparent area representing $\mu_{CSCS,TOT} \pm \sigma_{CSCS,TOT}$ and the white transparent area representing the [min;max] of the ensemble of CSCS trajectories, in PgC, right y-axis) and the relative impact on the CSCS dispersion of the three drivers ($r_{CSCS,D}$, in percentage, left y-axis, with D being CCO2 (in blue), LUC (in orange) or NIN (in green)).**


## Code availability

The source code of the ORCHIDEE-v3 model used in in this study is freely available online (DOI: 10.14768/9af22472-c438-41d7-815e-09d629e55cf8)

## Author contributions

NV designed the study; JARS performed the simulations, processed the data and created the visualizations; all authors contributed to the analysis; NV drafted the manuscript with contributions from JARS and PP; all authors reviewed and edited the manuscript.

**Competing interests**

The authors declare that they have no conflict of interest.

**Acknowledgements**

This project has received funding from the European Union's Horizon 2020 research and innovation programme under Grant Agreement N° 101003536 (ESM2025 – Earth System Models for the Future). This work was granted access to the HPC resources of GENCI-TGCC under the allocation A0130106328. JARS acknowledges for support from the Commissariat à

l'Energie Atomique et aux Energies Alternatives (CFR grant).

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

| Simulation | SSP | | | | | | | | |
|---|---|---|---|---|---|---|---|---|---|
| | **Hist** | **1-1.9** | **1-2.6** | **2-4.5** | **3-7.0** | **4-3.4** | **4-6.0** | **5-3.4os** | **5-8.5** |
| Marker | -7.7 | 58.6 | 83.1 | 103.8 | 86.9 | -5.6 | 71.0 | 75.8 | 115.5 |
| NIN sensitivity | / | 74.1±12.2 | / | / | 70.7±13.6 | -1.1±10.9 | / | / | 111.1±13.5 |
| LUC sensitivity | / | 11.66±38.1 | / | / | 70.4±44.5 | 30.0±40.3 | / | / | 78.9±46.2 |
| LUC + NIN sensitivity | / | 24.9±37.2 | / | / | 86.5±43.6 | 47.1±39.3 | / | / | 95.7±45.3 |

**Table 1: Change in land carbon store (PgC) for the historical period from 1850 to 2015 (Hist) and for the SSPs from 1850 to 2100 by using the marker simulation (Marker) or an ensemble of simulations with different nitrogen deposition trajectories and fertilisation (NIN sensitivity, $\mu^s_{CLCS,NIN} \pm \sigma^s_{CLCS,NIN}$, eq. 5), different land-use change trajectories (LUC sensitivity, $\mu^s_{CLCS,LUC} \pm$**
 **$\sigma^s_{CLCS,LUC}$, eq. 4) or different LUC and NIN trajectories (LUC + NIN sensitivity, $\mu^s_{CLCS,LUC+NIN} \pm \sigma^s_{CLCS,LUC+NIN}$, eq. 6). Positive values indicate a gain of carbon in the land reservoir.**

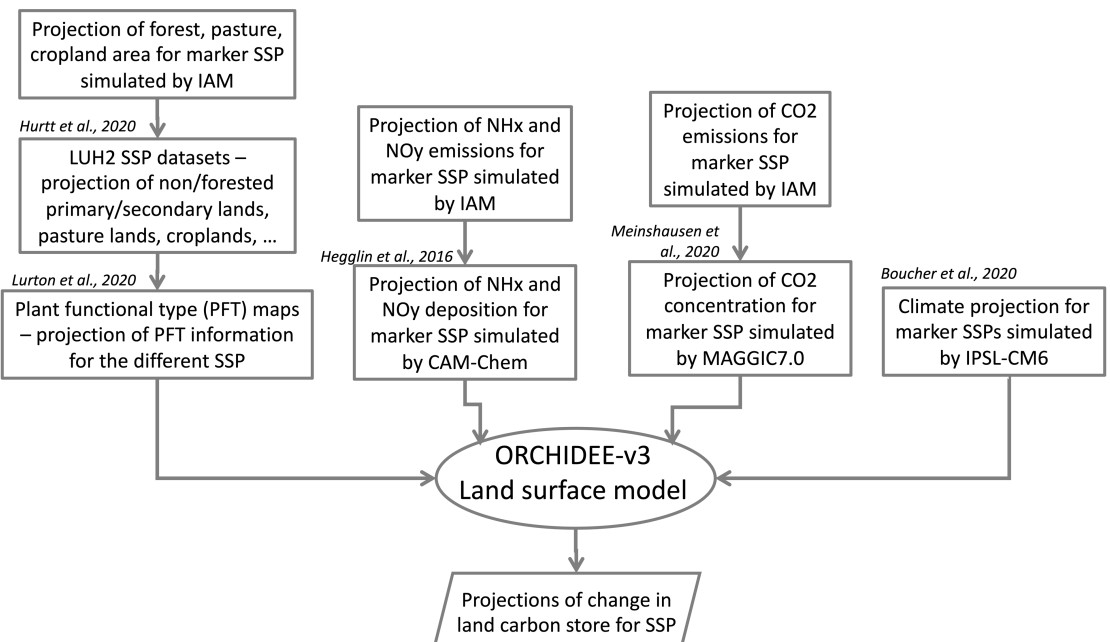

585 **Figure 1: Flow chart of the modelling framework highlighting the different input data (rectangles), the land surface model (ellipsoid) used in this study and the main output data produced (parallelogram)**

## Global forested land area

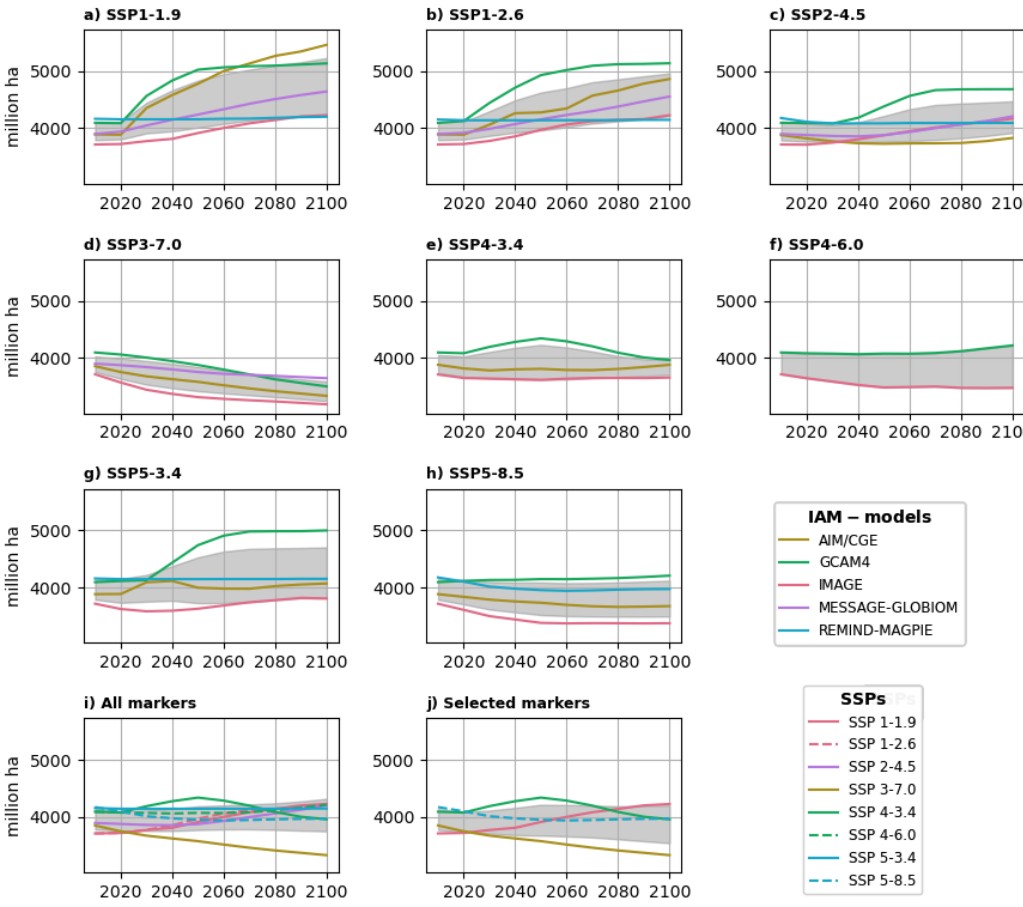

**Figure 2: Time evolution (2015-2100) of the global forested land area (Mha) projected by (a to h) different Integrated Assessment Models (IAM) for different Shared Socio-economic Pathways, (i) all IAM markers and (j) the selected IAM markers used in the study. Grey aeras represent the time evolution of the mean ± sigma.**

**Global NHx emissions**

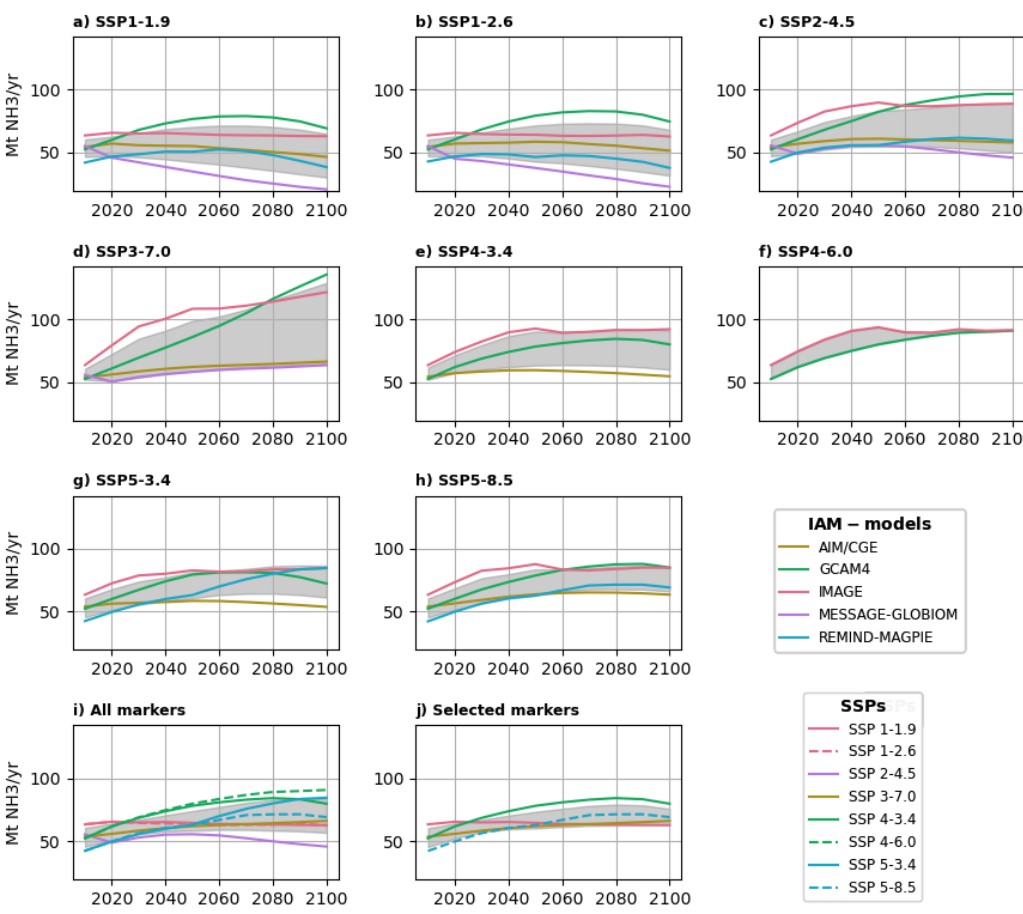

**Figure 3:** Time evolution (2015-2100) of the global NHx (NH3) emissions (Mt(NH3) yr[-1]) projected by (a to h) different Integrated Assessment Models (IAM) for different Shared Socio-economic Pathways, (i) all IAM markers and (j) the selected IAM markers used in the study. Grey aeras represent the time evolution of the mean ± sigma.

**Change in global land carbon store**

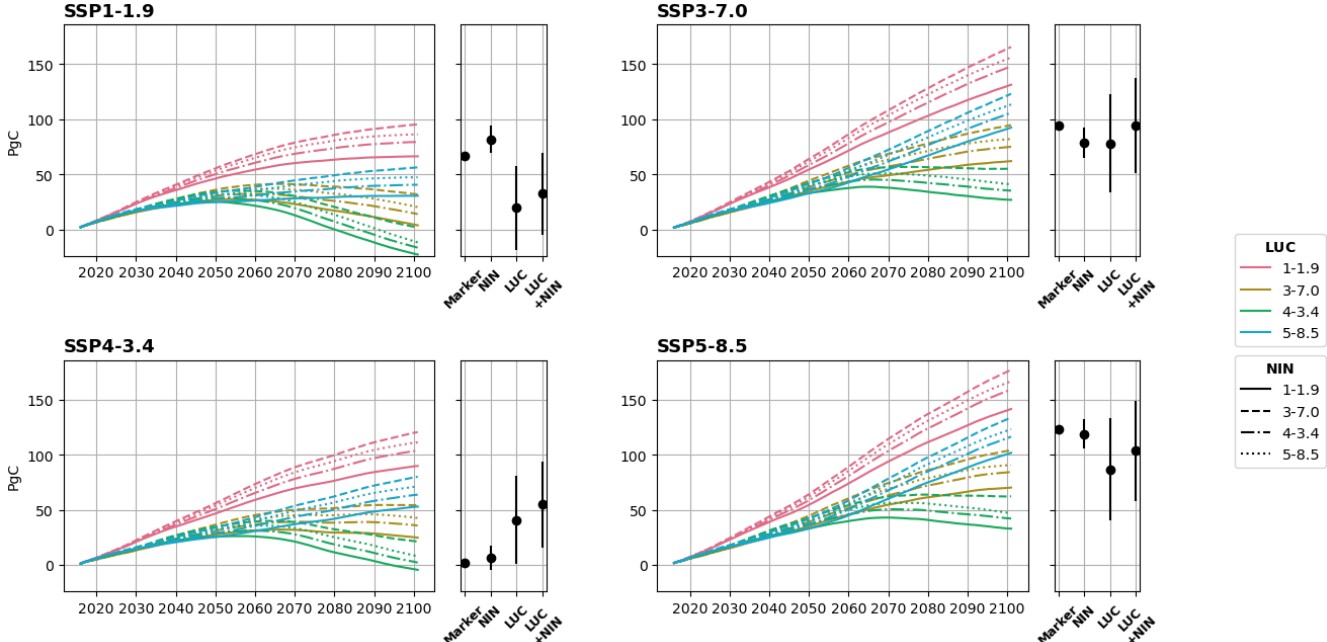

Figure 4: **Time evolution over 2015-2100 (left-side plot of each subpanel) of the global change in land carbon store (CLCS, PgC) driven by the four atmospheric [CO$_2$] and associated climate trajectories of the selected SSPs (subpanels SSP1-1.9, SSP3-7.0, SSP4-3.4 and SSP5-8.5) and by different trajectories for land-use change (LUC sensitivity; pink, brown, green and blue lines for SSP 1-1.9, 3-7.0, 4-3.4 and 5-8.5, respectively) and nitrogen deposition and fertilisation (NIN sensitivity; solid, dashed, dash-dotted and dotted lines for SSP 1-1.9, 3-7.0, 4-3.4 and 5-8.5, respectively). Right-side plot of each subpanel represents CLCS in 2100 by using the marker simulation (Marker), or an ensemble of simulations with different nitrogen deposition and fertilisation trajectories (NIN, $\mu^s_{CLCS,NIN} \pm \sigma^s_{CLCS,NIN}$, eq. 5), different land-use change trajectories (LUC, $\mu^s_{CLCS,LUC} \pm \sigma^s_{CLCS,LUC}$, eq. 4) and different LUC and NIN trajectories (LUC + NIN, $\mu^s_{CLCS,LUC+NIN} \pm \sigma^s_{CLCS,LUC+NIN}$, eq. 6)**

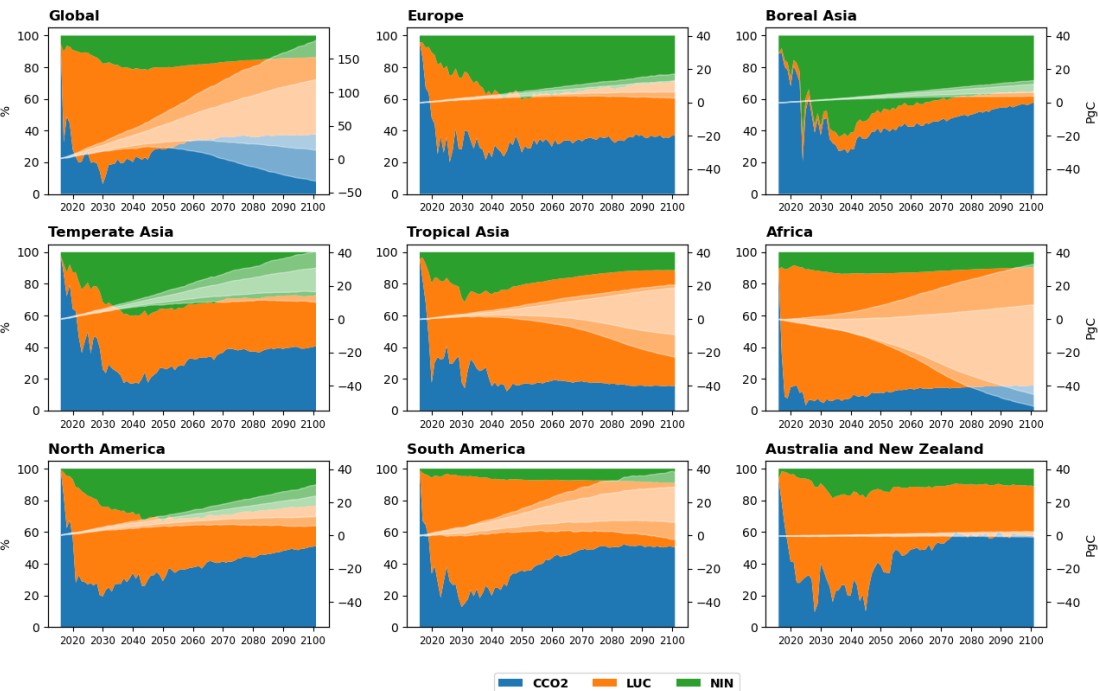

610 **Figure 5: Time evolution (2015-2100) of the change in land carbon store accounting for different atmospheric [CO₂] and associated climate (CCO2), land-use change (LUC) and atmospheric N deposition and fertilisation (NIN) trajectories (with the white semi-transparent area representing $\mu_{CLCS,TOT} \pm \sigma_{CLCS,TOT}$ (eq. 7) and the white transparent area representing the [min;max] of the ensemble of CLCS trajectories, in PgC, right y-axis) and the relative impact on the CLCS dispersion of the three drivers ($r_{CLCS,D}$ (eq. 11), in percentage, left y-axis, with D being CCO2 (in blue), LUC (in orange) or NIN (in green)).**

615

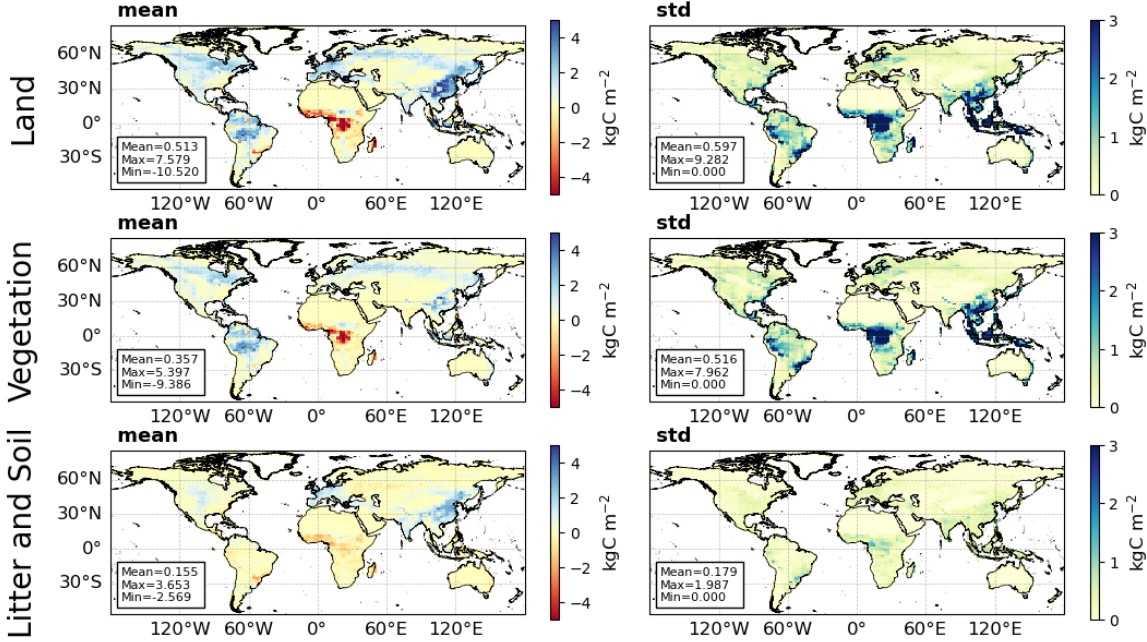

**Figure 6: Mean ($\mu_{CLCS,TOT}$) and standard deviation ($\sigma_{CLCS,TOT}$) of the change by 2100 (relatively to 2014) in carbon stored in land (CLCS), vegetation (CVCS) and litter+soil (CSCS) accounting for all the different trajectories regarding atmospheric [CO$_2$] and associated climate (CCO2), land-use change (LUC) and atmospheric N deposition and fertilisation (NIN)**

620

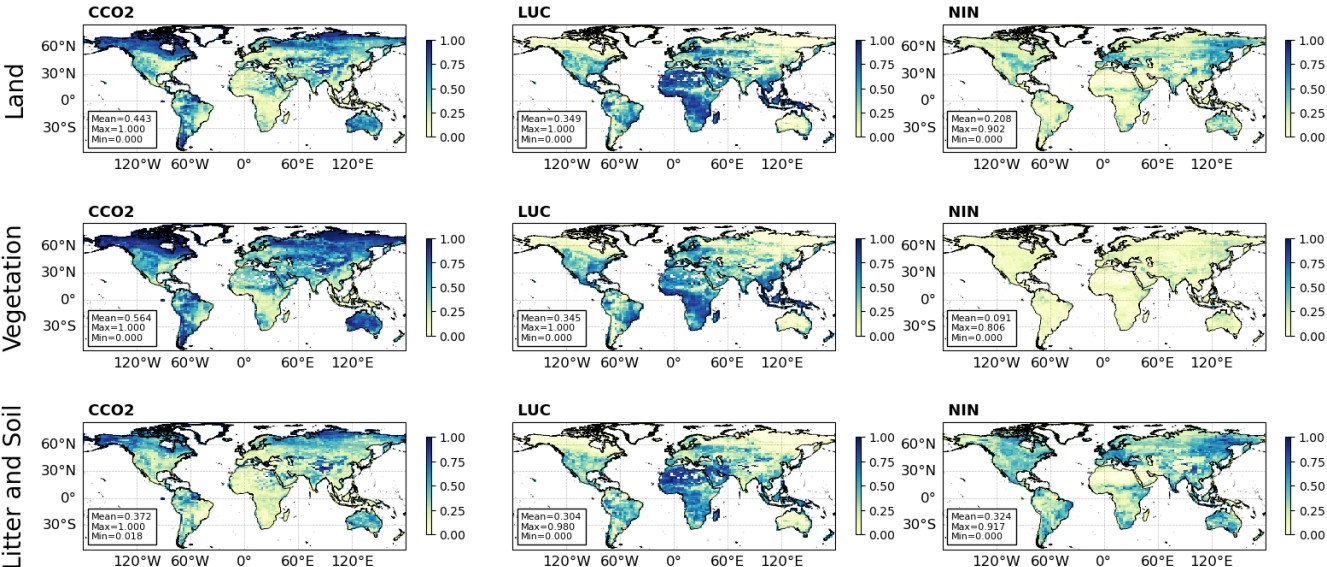

**Figure 7: Relative impact ($r_{CLCS,D}$ (eq. 11)) of the different trajectories regarding atmospheric [CO₂] and associated climate (CCO2), land-use change (LUC) and atmospheric N deposition on the change by 2100 (relatively to 2014) in carbon stored in land (CLCS), vegetation (CVCS) and litter+soil (CSCS)**

625

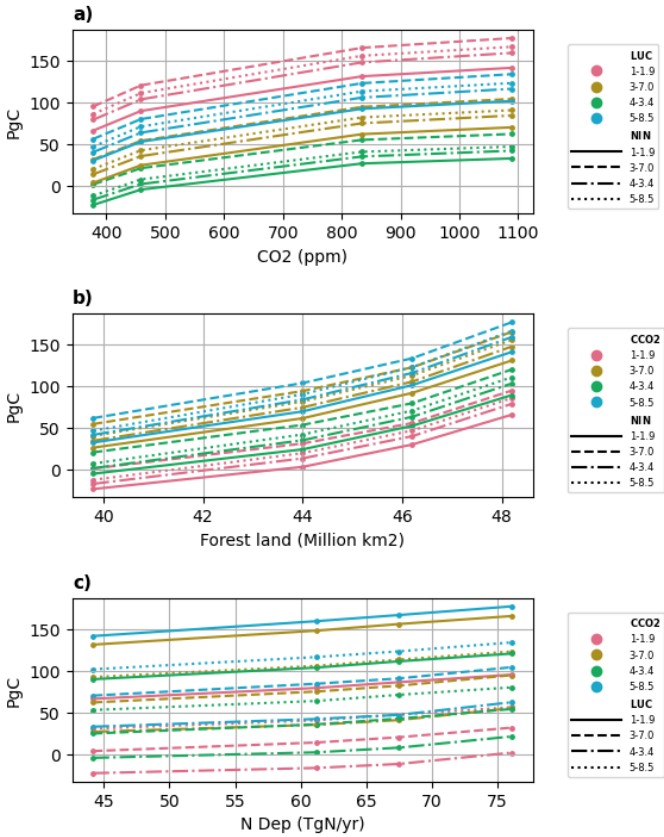

**Figure 8: CLCS in 2100 as a function of one of the studied drivers (i.e., a) atmospheric CO$_2$ level for CCO2, b) Forested lands for LUC and c) Atmospheric N deposition for NIN in 2100) for an ensemble of sixteen simulations driven by the different combinations of the two other drivers.**

630