# Peer review of "Projected changes in land carbon store over the 21st century: what contributions from land-use change and atmospheric nitrogen deposition?"

_Earth System Dynamics, 2023_

## Referee Comment (RC2)

**Review for "Projected changes in land carbon store over the 21st century: what contributions from land-use change and atmospheric nitrogen deposition?"**

Summary- In this study, the authors present an analysis of land carbon store (or CLCS) for different scenarios where different variables projected by Integrated Assessment Models (IAMs) are used as forcings for a land surface model – ORCHIDEE-v3. Specifically, the authors present results for variation in the CLCS results associated with changing values of CO2 concentrations , changing values of land use and changing values of N deposition. The authors have found that there is significant regional heterogeneity when it comes to the sensitivity of CLCS change to change in aforementioned factors.

Coupling of IAMs and ESMs is a topic of rising importance and is indeed gaining more attention as a part of the CMIP exercises. Moreover, the projections of variables such as nitrogen deposition in IAMs are relatively under studied and are therefore important to quantify via coupling with ESMs. Such coupling exercises can indicate what IAMs are missing. Therefore, this is clearly an important topic and an important question.

I largely followed the paper and its findings. However, I had some questions regarding the findings themselves and regarding the methodology. I recommend publication after the authors respond to the comments-

**Main comments**

1) **Inter SSP spread vs inter IAM spread-** I understand that the authors have used the "marker" scenarios for selected SSP scenarios as the inputs to the ESM since those are the only ones available. However, I'm not sure about the places where the authors conclude that the Inter SSP spread for the marker scenarios is similar to the inter IAM spread. The Inter IAM spread is largely the result of parameterizations and modelling choices (e.g., AIM is a CGE model while GCAM is a partial equilibrium model). The SSPs are socio-economic storylines on the other hand. Comparing the two seems like an apples to oranges question to me. I agree that the authors have concluded by saying more scenarios need to be made available (other than these marker scenarios). However, it still seems unconvincing to me to treat the marker spread as the IAM spread. Also note that the way land use, nitrogen and carbon cycle is modelled may be very different from IAM to IAM. Can the authors produce the spread for a region or two to assess the robustness of their assumption (as opposed to the global spread for the selected variables)?

2) **Documentation of IAM processes-** I believe this paper would benefit by a table which documents which IAMs are used in which marker scenarios and a summary of the assumptions used by the IAM for the land use change and N deposition modelling. The description does not have to be extensive, and the idea here would be that the reader would know what overall assumptions are going into these marker scenarios for the selected variables.

3) **PFT driven differences-** The other reviewer alluded to this as well, but it seems that there may be fundamental land type (or PFT)driven differences across scenarios. Can the authors document how the CLCS from individual or aggregated land types looks across scenarios? Can the authors also add a figure which shows the responses across different land types?

4) **Spatial results-** The regional heterogeneity is indeed interesting. I wanted to know if it was possible to show the mu or sigma values calculated as a map to identify hotspots for different variables. Basically, Figure A11 shown as a map with 3 facets (CCO2, LUC and NIN). This would really be an interesting analysis and also help pull out some within region dynamics. I also believe this is one of the bigger advantages of this coupling exercise.

5) **Takeaways for IAM modelers-** I apologize if this sounds vague. But can the authors frame some takeaways for IAM modelers other than the important point that more IAM scenarios need to be made available at a fine resolution? Sensitivity analysis such as these are often used to indicate areas where IAMs are weak and should produce better results or future focus areas for IAMs. Can the authors broaden the discussion to include some takeaways? One obvious one is that modelling of nitrogen deposition can have a significant impact on CLCS storage in some regions. Perhaps there are few more points that can be used.

6) **Description of methodology-** This manuscript would benefit from the inclusion of a flow chart which shows the inputs and the outputs. For example, IAM marker scenarios are inputs to the ORCHIDEE model. Also, there is a step where the LUH2 data (IAM marker scenarios) are further transformed to match ORCHIDEE's PFTs, correct? Can this be described in more detail? Did this downscaling add more uncertainty?

**Minor points**

1) **Page 1 Line 14-15-** This is a bit awkwardly worded. Perhaps you can cut the sentence at " More precisely, only one IAM output is used as representative of a single SSP". I'm not sure what the rest of the sentence adds.

2) **Page 2 Line 34-35-** " In the following, and by simplicity, we refer to these eight scenarios as SSPs" can be " Here forward we refer to these scenarios as SSPs for simplicity."

3) **Page 3 Line 64-66-** Note that recently there have been two way coupling exercises to couple IAMs and LSMs to address such uncertainties. See the E3SM exercises (https://agupubs.onlinelibrary.wiley.com/doi/full/10.1029/2022MS003156) as an example. This can probably be cited to ground the current study better. Note I am no way related to the study mentioned here!

4) **Page 3 Line 77-78-** Is ORCHIDEE a part of the Global Carbon Project suite of models? I see that the Fridgelstein paper is cited later, but perhaps that can be explicitly mentioned as well (If that is true).

5) **Page 3 Line 91-92-** What resolution does ORCHIDEE operate at? Is it the same resolution as LUH2 or is it something different? That can be mentioned somewhere.

---

## Author Comment (AC1)

We thank the reviewer #1 for the constructive comments on our manuscript. Below, we answer to each of them. In black, are the reviewer comments and in red our responses and in italics, the additional text we propose to add in the revised manuscript.

In the manuscript "Projected changes in land carbon store over the 21st century: what contributions from land-use change and atmospheric nitrogen deposition?", the authors evaluated the changes of CLCS projections in SSPs, focusing the uncertainties from land use change and nitrogen deposition. The authors concluded that the projection spread contributed by land use change spread is larger than from CO2 spread, and the nitrogen deposition has relatively smaller contribution. This conclusion is important and has significant implications for understanding the projections of the Earth system models. However, I think the presentation of this manuscript need to be improved and a few concerns need to be addressed before publishing.

– The main concern I have is about the land use change and N deposition spread. The authors "used the selected SSP markers spread as a proxy for the inter-IAM spread". Although the authors compared the two spreads, and showed them to have similar magnitude. There is still possibility that the differences of vegetation types are systematically different among SSP markers and IAMs. i.e. the SSP markers spread could come from the scenario differences while inter-IAM spread for the same scenario is from model differences. The same vegetation type distributed in different regions (e.g. tropical and boreal forests) may have different C source/sink. Therefore, besides the global analyses, it is necessary to check where the differences are in SSP markers spread and inter-IAM spread.

We thank the reviewer #1 for the suggestion of performing a regional analysis. Such regional analysis will help to ensure that the SSP markers spread is comparable to the inter-IAM spread not only at global scale but also at regional scale. The data we used for processing Figures 1, 2 but also A1, A2 and A3 is IAM output data produced for CMIP6, available on the SSP Database (https://tntcat.iiasa.ac.at/SspD). They are accessible at global scale but also for five aggregated geographical and/or economical regions. These five regions are "Asia" (ASIA), "Latin America" (LAM), "Reforming economies (REF), "Middle East and Africa" (MEA) and countries from the "Organisation for Economic Co-operation and Development" (OECD). We re-processed for the five aggregated regions (see Figures below) the Figure 1 which shows the time evolution (2015-2100) of the forested land area projected by different Integrated Assessment Models (IAM) for different Shared Socio-economic Pathways. The results of this regional analysis show that the inter-IAM spread is significant for any of the five regions and comparable to the selected SSP markers spread. As a consequence, the assumption of using the selected SSP markers spread as a proxy for the inter-IAM spread based on a global analysis remains valid at regional scale. We do not suggest to keep these extra figures as part of the manuscript nor in the Appendix. Nevertheless, we propose to add extra information in the manuscript reporting on this regional analysis at line 130:

*"The comparison between inter-SSP markers and inter-IAM trajectories for the different SSPs is presented at global scale, but the conclusion that the selected SSP markers*

*spread is comparable to the inter-IAM spread for the different SSPs remains valid at regional scale (based on the data available on the SSP Database for five aggregated regions (*"Asia", "Latin America", "Reforming economies", "Middle East and Africa" and countries from the "Organisation for Economic Co-operation and Development"*), not shown).*

[Figure]

Forested land area - REF region

**Forested land area - LAM region**

[Figure]

**Forested land area - ASIA region**

[Figure]

a) SSP1-1.9

b) SSP1-2.6

c) SSP2-4.5

d) SSP3-7.0

e) SSP4-3.4

f) SSP4-6.0

g) SSP5-3.4

h) SSP5-8.5

i) All markers

j) Selected markers

**IAM − models**
- AIM/CGE
- GCAM4
- IMAGE
- MESSAGE-GLOBIOM
- REMIND-MAGPIE

**SSPs**
- SSP 1-1.9
- SSP 1-2.6
- SSP 2-4.5
- SSP 3-7.0
- SSP 4-3.4
- SSP 4-6.0
- SSP 5-3.4
- SSP 5-8.5

**Forested land area - MAF region**

[Figure]

[Figure]

**Forested land area - OECD region**

– Also, I think the authors need to carefully use the term "uncertainty". In my opinion, the forcings are from different scenarios and the differences due to the forcings are thus not something "uncertain" but some "certain" signal. The use of "spread" is also somehow inaccurate as we usually use the term for "model spread". I prefer to explain the results as scenario difference rather than "uncertainty".

We agree that the use of the term "uncertainty" may not be always appropriated in the manuscript. Indeed, we used the term "uncertainty" when referring to uncertainty on land-use change or nitrogen deposition trajectories for instance because ultimately that is the uncertainty from the different IAMs we would like assess. Unfortunately, because these IAM trajectories for the different SSPs are not gridded and harmonized, we used the selected SSP markers spread as a proxy for the inter-IAM. You are right that the different SSP trajectories do not strictly reflect a model uncertainty but indeed more "differences" obtained from different assumption in terms of socio-economic development and mitigation target.

We thus propose to change a much as possible the word "uncertainty", when it refers directly to various SSPs, to "differences". As an example:

The Initial sentence in the abstract: "Through a set of land-only factorial simulations, we specifically aim at estimating the **CLCS uncertainties** associated with land-use change and nitrogen deposition trajectories."

New sentence: *"Through a set of land-only factorial simulations, we specifically aim at estimating the CLCS **differences** associated with **differences in** land-use change and nitrogen deposition trajectories."*

With respect to differences induced by IAM model structure, we choose to keep in certain cases the term "model uncertainties" and to be more explicit with the terms "model spread" or "model differences". Note that for the land surface model evaluation (TRENDY inter-comparison) the term model uncertainties is often used to qualify some outputs.

However, in some particular cases it is more appropriate to keep the word uncertainty, although it also partly refers to different assumptions. We thus also added the following sentence, line 115: *"Given that, ultimately, we would like to assess the uncertainty associated to land-use and nitrogen inputs from the different IAMs for any SSP, in the following we may use the term "uncertainty" when referring to the different inter-SSP markers trajectories although they correspond to more certain trajectories obtained for different assumptions in terms of socio-economic development and mitigation level."*

Finally with respect to the term "spread" we do not agree with the reviewer as indeed it is often used as "model spread" but we believe it is more general and can thus be used in other context to refer to differences. We have however tried to restrict a bit its use throughout the revised manuscript.

– My second concern is that this study is based on a single model ORCHIDEE-v3. Given the large differences among land surface models, I wonder how robust the results are. I am not asking to add new simulations, but discussing this uncertainty is helpful. The authors may compare ORCHIDEE-v3 and other models' performances in the TENDY land use change experiments to estimate the robustness of this study.

We fully agree with this comment. The end of the Discussion section is referring to this topic. In particular, line 284, there is the following sentence: "This limited set of studies thus highlights the need of performing the multi-sensitivity analysis we proposed in this paper with an extended ensemble of models, in order to evaluate how our conclusions can be shared across models with different representations of the key C-related ecosystem processes."

As suggested by the reviewer, we propose to add some information about the ORCHIDEE-v3 model performances within the TRENDY ensemble. O'Sullivan et al. (2022) developed an extended multi-model analysis of the drivers of the land carbon sink and its sources of uncertainty based on the TRENDY models ensemble. Their figure 3 shows the multi-model ensemble of the time evolution over the last six decades of the change in carbon stored in the vegetation and in the soil pools due to change in CO2 and

Nitrogen deposition, climate, and land use change. The Figure 2 in the supplementary information of O'Sullivan et al. paper shows the same information for each of the eighteen models of the TRENDY ensemble, including ORCHIDEE-v3. This figure highlights the high inter-model dispersion for any of the six trajectories in terms of carbon changes (2 pools x 3 drivers). The comparison of the two figures shows that the changes in carbon stored in vegetation and soil due to the three main drivers as simulated by the ORCHIDEE-v3 model are very similar to those computed as the multi model ensemble mean. The only significant difference is obtained for the change in carbon stored in the soil pool due to land-use change: the multi-model ensemble mean estimates a loss of carbon in soil due to land-use changes of about 25 PgC between 1960 and 2020 while ORCHIDEE-v3 estimates there is no change by 2020. Note that this difference of ~25 PgC remains in the ±1sigma interval of the TREND models distribution (see Figure 3b of O'Sullivan et al., 2022).

We propose to rewrite and extend the sentence lines 284-286 ("This limited set of studies thus highlights the need of performing the multi-sensitivity analysis we proposed in this paper with an extended ensemble of models, in order to evaluate how our conclusions can be shared across models with different representations of the key C-related ecosystem processes.") as follows:

*"The large inter-model spread regarding changes in land carbon store has also been reported in many studies such as the one of Liddicoat et al. (2021) based on the CMIP6 historical and SSPs experiments or the one of O'Sullivan et al. (2022) based on the TRENDY land models ensemble over the last six decades. In this latter study, eighteen land surface models were used to assess the changes in carbon stored in vegetation and soil due to change in CO2 and Nitrogen deposition, climate, and land use. ORCHIDEE-v3 was one of these models and showed results very similar to those obtained with the multi model ensemble means which gives confidence on how relevant are the results of the present study. Nevertheless, there is a need of performing the multi-sensitivity analysis we proposed in this paper with an extended ensemble of models, in order to evaluate how our conclusions can be shared across models with different representations of the key C-related ecosystem processes."*

– Finally, I suggest the authors to divide Section 3 into subsections, so the readers can better follow and capture the key points.

Thank you for this suggestion. We propose the following subsection titles:

"*Change in land carbon store (CLCS) over the historical period and for the different SSPs experiments*": From line 173 to 197

"*Spatial and temporal analysis of the CLCS dispersion and its drivers*": From line 198 to 227

"*Change in carbon stored in vegetation and litter and soil pools*": From line 228 to 248

*"CLCS as a function of atmospheric CO2, Forested land area and atmospheric nitrogen deposition"*: From line 249 to 269

*"Comparison with other studies and path for future research"*: From line 270 to 286

– Line 57: non-makers -> non-markers

This will be corrected for in the revised manuscript

---

## Author Comment (AC2)

We thank the reviewer #2 for the constructive comments on our manuscript. Below, we answer to each of them. In black, are the reviewer comments and in red our responses and in italics, the additional text we propose to add in the revised manuscript.

Summary

In this study, the authors present an analysis of land carbon store (or CLCS) for different scenarios where different variables projected by Integrated Assessment Models (IAMs) are used as forcings for a land surface model – ORCHIDEE-v3. Specifically, the authors present results for variation in the CLCS results associated with changing values of $CO_2$ concentrations, changing values of land use and changing values of N deposition. The authors have found that there is significant regional heterogeneity when it comes to the sensitivity of CLCS change to change in aforementioned factors.

Coupling of IAMs and ESMs is a topic of rising importance and is indeed gaining more attention as a part of the CMIP exercises. Moreover, the projections of variables such as nitrogen deposition in IAMs are relatively under studied and are therefore important to quantify via coupling with ESMs. Such coupling exercises can indicate what IAMs are missing. Therefore, this is clearly an important topic and an important question.

I largely followed the paper and its findings. However, I had some questions regarding the findings themselves and regarding the methodology. I recommend publication after the authors respond to the comments-

Main comments

1) Inter SSP spread vs inter IAM spread- I understand that the authors have used the "marker" scenarios for selected SSP scenarios as the inputs to the ESM since those are the only ones available. However, I'm not sure about the places where the authors conclude that the Inter SSP spread for the marker scenarios is similar to the inter IAM spread. The Inter IAM spread is largely the result of parameterizations and modelling choices (e.g., AIM is a CGE model while GCAM is a partial equilibrium model). The SSPs are socio-economic storylines on the other hand. Comparing the two seems like an apples to oranges question to me. I agree that the authors have concluded by saying more scenarios need to be made available (other than these marker scenarios). However, it still seems unconvincing to me to treat the marker spread as the IAM spread. Also note that the way land use, nitrogen and carbon cycle is modelled may be very different from IAM to IAM. Can the authors produce the spread for a region or two to assess the robustness of their assumption (as opposed to the global spread for the selected variables)?

We agree on the "conceptual" difference between the inter IAM spread and inter SSP spread for the marker. Yes, the IAM spread is largely induced by different modelling frameworks, while the SSP spread correspond to different socio-economic storylines. What we wanted to highlight is that, although they have different origins, these two spreads are of similar magnitude when looking at IAM's variables used directly or

indirectly to constrain land surface models. In the manuscript, we showed comparisons between inter-IAM spread and inter-SSP spread at global scale. In order to ensure that what has been highlighted at global scale remains valid at regional scale, we performed a regional analysis, as suggested by the reviewer. The data we used for processing Figures 1, 2 but also A1, A2 and A3 is IAM output data produced for CMIP6, available on the SSP Database (https://tntcat.iiasa.ac.at/SspD). They are accessible at global scale but also for five aggregated geographical and/or economical regions. These five regions are "Asia" (ASIA), "Latin America" (LAM), "Reforming economies (REF), "Middle East and Africa" (MEA) and countries from the "Organisation for Economic Co-operation and Development" (OECD). We re-processed for the five aggregated regions (see Figures below) the Figure 1 which shows the time evolution (2015-2100) of the forested land area projected by different Integrated Assessment Models (IAM) for different Shared Socio-economic Pathways. The results of this regional analysis show that the inter-IAM spread is significant for any of the five regions and comparable to the selected SSP markers spread. As a consequence, the assumption of using the selected SSP markers spread as a proxy for the inter-IAM spread based on a global analysis remains valid at regional scale. We do not suggest to keep these extra figures as part of the manuscript nor in the Appendix. Nevertheless, we propose to add extra information in the manuscript reporting on this regional analysis.

Line 130, we propose to add the following sentence:

*"The comparison between inter-SSP markers and inter-IAM trajectories for the different SSPs is presented at global scale, but the conclusion that the selected SSP markers spread is comparable to the inter-IAM spread for the different SSPs remains valid at regional scale (based on the data available on the SSP Database for five aggregated regions (*"Asia", "Latin America", "Reforming economies", "Middle East and Africa" and countries from the "Organisation for Economic Co-operation and Development"*), not shown).*

**Forested land area - REF region**

[Figure]

**a)** SSP1-1.9
**b)** SSP1-2.6
**c)** SSP2-4.5
**d)** SSP3-7.0
**e)** SSP4-3.4
**f)** SSP4-6.0
**g)** SSP5-3.4
**h)** SSP5-8.5
**i)** All markers
**j)** Selected markers

**IAM − models**
- AIM/CGE
- GCAM4
- IMAGE
- MESSAGE-GLOBIOM
- REMIND-MAGPIE

**SSPs**
- SSP 1-1.9
- SSP 1-2.6
- SSP 2-4.5
- SSP 3-7.0
- SSP 4-3.4
- SSP 4-6.0
- SSP 5-3.4
- SSP 5-8.5

**Forested land area - LAM region**

[Figure]

**Forested land area - ASIA region**

[Figure]

**Forested land area - MAF region**

[Figure]

[Figure]

**Forested land area - OECD region**

2) Documentation of IAM processes- I believe this paper would benefit by a table which documents which IAMs are used in which marker scenarios and a summary of the assumptions used by the IAM for the land use change and N deposition modelling. The description does not have to be extensive, and the idea here would be that the reader would know what overall assumptions are going into these marker scenarios for the selected variables.

We thank reviewer #2 for the suggestion. Indeed, adding a table with information on the IAMs and the assumptions done about land modelling will be useful. We propose to prepare and add such table in the revised manuscript based on the information we can gather from https://www.iamcdocumentation.eu/index.php/IAMC_wiki#Documentation

3) PFT driven differences- The other reviewer alluded to this as well, but it seems that there may be fundamental land type (or PFT) driven differences across scenarios. Can the authors document how the CLCS from individual or aggregated land types looks

across scenarios? Can the authors also add a figure which shows the responses across different land types?

Thanks for this suggestion. We propose to add three additional figures as Supplementary information in the revised manuscript, similar to the original Figure 4 but focusing respectively on forested lands, grasslands and croplands. Nevertheless, in order to avoid that the signals represented in these figures are impacted by the change in areas for respectively forest lands, grasslands, croplands, due to land-use changes, we propose to not express the CLCS in 'absolute values' (in PgC) but rather per unit area of respectively forest lands, grasslands and croplands (expressed in kgC m$^{-2}$). These figures are not presented here as their production requires to get access to output data per vegetation type (PFTs) and specific processing (a longer than initially expected process), but they will be included and discussed in the revised version of the manuscript.

4) Spatial results- The regional heterogeneity is indeed interesting. I wanted to know if it was possible to show the mu or sigma values calculated as a map to identify hotspots for different variables. Basically, Figure A11 shown as a map with 3 facets (CCO2, LUC and NIN). This would really be an interesting analysis and also help pull out some within region dynamics. I also believe this is one of the bigger advantages of this coupling exercise.

Yes, indeed, it's a good suggestion. We processed the suggested figures which are shown here below. The figure A shows the mean ($\mu_{CLCS,TOT}$) and standard deviation ($\sigma_{CLCS,TOT}$) of the change in carbon by 2100 (relatively to 2014) accounting for all the different CCO2, LUC and NIN trajectories for the total land (CLCS), vegetation (CVCS) and litter+soil (CSCS) reservoirs. These maps correspond to the spatial analysis of the information represented by the white area in 2100 on Figure 4, Figure A11 and Figure A12. The figure B represents the relative impact by 2100 on the CLCS, CVCS and CSCS dispersions of the three drivers (ie. CCO2, LUC and NIN). It corresponds to the spatial analysis of the $r_{CVCS,D}$ variable shown on Figure 4, Figure A11 and Figure A12 with the blue, orange and green stacks.

We propose to add these two additional figures in the core of the manuscript with relevant description of the spatial distribution of each diagnostic.

**Change in carbon store over 2015-2100**

[Figure]

*Figure A - Mean ($\mu_{CLCS,TOT}$) and standard deviation ($\sigma_{CLCS,TOT}$) of the change in carbon by 2100 (relatively to 2014) stored in land (CLCS), vegetation (CVCS) and litter+soil (CSCS) accounting for all the different trajectories regarding atmospheric [CO2] and associated climate (CCO2), land-use change (LUC) and atmospheric N deposition and fertilisation (NIN)*

**Relative impacts on the change in carbon store**

[Figure]

*Figure B - Relative impact ($r_{CLCS,D}$ (eq. 11)) of the different trajectories regarding atmospheric [CO2] and associated climate (CCO2), land-use change (LUC) and atmospheric N deposition on the change in carbon by 2100 (relatively to 2014) stored in land (CLCS), vegetation (CVCS) and litter+soil (CSCS)*

5) Takeaways for IAM modelers- I apologize if this sounds vague. But can the authors frame some takeaways for IAM modelers other than the important point that more IAM scenarios need to be made available at a fine resolution? Sensitivity analysis such as these are often used to indicate areas where IAMs are weak and should produce better results or future focus areas for IAMs. Can the authors broaden the discussion to include some takeaways? One obvious one is that modelling of nitrogen deposition can have a significant impact on CLCS storage in some regions. Perhaps there are few more points that can be used.

We agree that the takeaways message for IAM modelers were relatively vague and not discussed enough in the original manuscript. We propose to add a few points in the "summary and conclusion" section at the end:

*"In addition, given the large impact of land use change differences between IAMs (for a given SSP) and the significant impact (although lower) of N inputs, we also recommend that the IAM community provides more information on the uncertainties associated to these drivers. For instance, it would be informative to obtain quantitative information on the uncertainty associated to these variables, with a high and a low range trajectory for each driver and whether these uncertainties stand from structural or parametric IAM uncertainties. Information on the degree of correlation between the uncertainty associated to each driver (land use and N inputs) would also help to propagate them in LSMs and ESMs simulations."*

6) Description of methodology- This manuscript would benefit from the inclusion of a flow chart which shows the inputs and the outputs. For example, IAM marker scenarios are inputs to the ORCHIDEE model. Also, there is a step where the LUH2 data (IAM marker scenarios) are further transformed to match ORCHIDEE's PFTs, correct? Can this be described in more detail? Did this downscaling add more uncertainty?

Thanks again for the suggestion. We will prepare a flow chart that will look like to the one here below. It is true that there is a step between the LUH2 data and the land-use maps used as input of ORCHIDEE. This procedure is very briefly mentioned in the manuscript at lines 95-96: "The procedure needed for translating the original data for land-use into the fifteen land classes of ORCHIDEE is described in Lurton et al. (2020)." We propose to add the following information: *"In this procedure, information regarding the cropland and pasture areas from LUH2 is preserved while natural land is split into the different unmanaged land classes of ORCHIDEE using data from the ESA CCI Land cover product (ESA, 2022)."*

This procedure is highly model-specific and, as a consequence, may add uncertainty when performing multi-model analysis. When using a single land surface model as we do in our study, we don't think this downscaling add significant uncertainty on the studied variables.

[Figure]

*Figure C – Flow chart of the modelling framework highlighting the different input data (rectangles), the land surface model (ellipsoid) used in this study and the main output data produced (parallelogram)*

Minor points

1) Page 1 Line 14-15- This is a bit awkwardly worded. Perhaps you can cut the sentence at " More precisely, only one IAM output is used as representative of a single SSP". I'm not sure what the rest of the sentence adds.

Thanks, the rest of the sentence will be removed.

2) Page 2 Line 34-35- " In the following, and by simplicity, we refer to these eight scenarios as SSPs" can be " Here forward we refer to these scenarios as SSPs for simplicity."

We'll rephrase the sentence as proposed. Thanks for the suggestion.

3) Page 3 Line 64-66- Note that recently there have been two way coupling exercises to couple IAMs and LSMs to address such uncertainties. See the E3SM exercises (https://agupubs.onlinelibrary.wiley.com/doi/full/10.1029/2022MS003156) as an example. This can probably be cited to ground the current study better. Note I am no way related to the study mentioned here!

We agree that indeed there are some attempts to have more coupled initiative linking IAMs and ESMs. The development of the E3ESM model represents one initiative from DOE (Department of Energy in the USA) to link energy questions to climate projections by ESMs. However, E3ESM only makes a first step in that direction as it does not yet

inlcude all economical drivers of the IAMs. We add the reference to Golaz et al., (2022) in addition to the one to Monier et al. (2018).

4) Page 3 Line 77-78- Is ORCHIDEE a part of the Global Carbon Project suite of models? I see that the Fridgelstein paper is cited later, but perhaps that can be explicitly mentioned as well (If that is true).

Yes, ORCHIDEE-v3 contributed to the Global Carbon Budget over the last four years. We rephrase the sentence at lines 87-88 to include this information: *"It also ranked with a good score for a set of key land variables in a recent model benchmark study (Seiler et al., 2022) as well as in the TRENDY model inter-comparison project **as part of the land surface models contributing to the Global Carbon Budget** (Friedlingstein et al., 2022)."*

5) Page 3 Line 91-92- What resolution does ORCHIDEE operate at? Is it the same resolution as LUH2 or is it something different? That can be mentioned somewhere.

No, indeed, ORCHIDEE runs at the resolution of the climate data, which is here the resolution of the IPSL-CM6 model (i.e. a global resolution of 2.5°x1.27° in longitude and latitude). We will add this information at line 101: *"In this study, ORCHIDEE-v3 ran at the same resolution as the climate input data (i.e. 2.5°x1.27°)."*

**References**

ESA: ESA CCI Land cover website, https://www.esa-landcover-cci.org/ (last access: 11 March 2022).

O'Sullivan, M., et al. (2022). "Process-oriented analysis of dominant sources of uncertainty in the land carbon sink." Nature Communications 13(1).

---

## Author Response (AR2)

As requested, we used only "Fig." to cite any figure in the manuscript except at line 102 where "Figure" is the beginning of a sentence (in line with the Copernicus recommendations for manuscript preparation). In addition, we modified the colour schemes used in the maps and charts in order to allow readers with colour vision deficiencies to correctly interpret them.